# A Benzodiazepine-Derived Molecule That Interferes with the Bio-Mechanical Properties of Glioblastoma-Astrocytoma Cells Altering Their Proliferation and Migration

**DOI:** 10.3390/ijms26062767

**Published:** 2025-03-19

**Authors:** Gregorio Ragazzini, Andrea Mescola, Riccardo Tassinari, Alessia Gallerani, Chiara Zannini, Domenico Di Rosa, Claudia Cavallini, Martina Marcuzzi, Valentina Taglioli, Beatrice Bighi, Roberta Ettari, Vincenzo Zappavigna, Carlo Ventura, Andrea Alessandrini, Lorenzo Corsi

**Affiliations:** 1Department of Physics, Informatics and Mathematics, University of Modena and Reggio Emilia, Via Campi 213/A, 41125 Modena, Italy; gregorio.ragazzini@unimore.it (G.R.); alessia.gallerani@unimore.it (A.G.); beatrice.bighi@unimore.it (B.B.); 2Eldor Lab, Via di Corticella 183, 40128 Bologna, Italy; riccardo.tassinari@eldorlab.com (R.T.); chiara.zannini@eldorlab.com (C.Z.); claudia.cavallini@eldorlab.com (C.C.); valentina.taglioli@eldorlab.com (V.T.); carlo.ventura@unibo.it (C.V.); 3CNR-Nanoscience Institute-S3, Via Campi 213/A, 41125 Modena, Italy; andrea.mescola@nano.cnr.it; 4Lab of Medicine and Genomics, Department of Medicine, Surgery and Dentistry “Scuola Medica Salernitana”, SaIA, University of Salerno, 84081 Baronissi, Italy; ddirosa@unisa.it; 5Department of Medical and Surgical Sciences, University of Bologna, Via Irnerio, 49, 40126 Bologna, Italy; martinamarcuzzi9@gmail.com; 6Department of Chemical, Biological, Pharmaceutical and Environmental Sciences, University of Messina, 98168 Messina, Italy; roberta.ettari@unime.it; 7Department of Life Sciences, University of Modena and Reggio Emilia, Via G. Campi 287, 41125 Modena, Italy; vincenzo.zappavigna@unimore.it; 8National Laboratory of Molecular Biology and Stem Cell Engineering, National Institute of Biostructures and Biosystems, Eldor Lab, Via di Corticella 183, 40128 Bologna, Italy

**Keywords:** GBM, invasion, microtubule, anticancer drug, biomechanics, spheroid

## Abstract

Glioblastoma multiforme (grade IV glioma) is characterized by a high invasive potential, making surgical intervention extremely challenging and patient survival very limited. Current pharmacological approaches show, at best, slight improvements in the therapy against this type of tumor. Microtubules are often the target of antitumoral drugs, and specific drugs affecting their dynamics by acting on microtubule-associated proteins (MAPs) without producing their depolymerization could affect both glioma cell migration/invasion and cell proliferation. Here, we analyzed on a cellular model of glioblastoma multiforme, the effect of a molecule (1-(4-amino-3,5-dimethylphenyl)-3,5-dihydro-7,8-ethylenedioxy-4h2,3-benzodiazepin-4-one, hereafter named 1g) which was shown to act as a cytostatic drug in other cell types by affecting microtubule dynamics. We found that the molecule acts also as a migration suppressor by inducing a loss of cell polarity. We characterized the mechanics of U87MG cell aggregates exposed to 1g by different biophysical techniques. We considered both 3D aggregates and 2D cell cultures, testing substrates of different stiffness. We established that this molecule produces a decrease of cell spheroid contractility and it impairs 3D cell invasion. At the same time, in the case of isolated cells, 1g selectively produces an almost instantaneous loss of cell polarity blocking migration and it also produces a disorganization of the mitotic spindle when cells reach mitosis, leading to frequent mitotic slippage events followed by cell death. We can state that the studied molecule produces similar effects to other molecules that are known to affect the dynamics of microtubules, but probably indirectly via microtubule-associated proteins (MAPs) and following different biochemical pathways. Consistently, we report evidence that, regarding its effect on cell morphology, this molecule shows a specificity for some cell types such as glioma cells. Interestingly, being a molecule derived from a benzodiazepine, the 1g chemical structure could allow this molecule to easily cross the blood–brain barrier. Thanks to its chemical/physical properties, the studied molecule could be a promising new drug for the specific treatment of GBM.

## 1. Introduction

Glioblastoma multiforme (GBM) is one of the most common malignant primary brain tumors and, despite numerous efforts to find an effective therapy, it is still characterized by a very poor prognosis [1,2]. Among others, one feature that makes GBM difficult to defeat is its ability to migrate and infiltrate into the surrounding brain parenchyma and these properties are strongly related to a continuous dynamic reorganization of the polymers making up the cytoskeleton of glioblastoma cells [3,4,5,6,7]. At the same time, many of the changes in cell properties during tumor progression induce significant alterations in the architecture and mechanical properties of both the tumor cells and the surrounding host tissue. The strategies against this type of tumor, like in other cases, are complicated by the potential go-or-grow behavior of the involved malignant cells. This mechanism refers to a hypothesis, derived mainly from in-vitro experiments, according to which the infiltrative and proliferative states of cancer cells are mutually exclusive phenotypes [8,9,10,11,12,13,14,15]. Moreover, a problem of glioma cells, which is also typical of other tumor cells, is their ability to adapt to changing environmental situations by adjusting, for example, the migration strategy to a variation of the physical environment [16,17,18]. Although the available standard therapy for GBM has evolved into multimodality approaches, comprising temozolomide and the antiangiogenic monoclonal recombinant antibody bevacizumab, corticosteroids, and immunotherapy, unfortunately, the improvements in patients’ survival have been only modest [19,20]. This has led to an important effort to identify new molecular targets in GBM tumors, specifically focusing on pathways involved in motility, applied traction force, and proliferative properties of the associated cells [21,22,23,24,25,26]. It is known that cellular and matrix mechanics are involved in many biological functions in eukaryotic cells, such as migration, differentiation, morphogenesis, and proliferation [27,28,29,30]. The cytoskeleton and the associated proteins are responsible both for transducing external mechanical stimuli into biochemical processes and for the application of stresses to the external matrix. This sort of mechanical reciprocity is strictly dependent on the activity of cytoskeleton-associated motor proteins and the cell/matrix adhesion molecules and it is altered in the case of developing and migrating tumors [31,32,33,34,35]. Specific microtubule-associated proteins (MAPs), especially those related to the highly dynamic behavior of microtubules or the ones connecting the cytoskeleton with the cortical layer of the cell, could represent a good focus to simultaneously act on the invasion and proliferation activity of GBM cells. From this point of view, compounds that selectively target MAPs inducing the alteration of microtubules dynamics but preventing their depolymerization, such as stabilizing drugs, could more selectively target cancer cells, in which the dynamic activity is strongly enhanced [36,37,38,39]. This is also interesting because it has been found that the increased or decreased phosphorylation of MAPs are related to the sensitivity of cancer cells to the use of drugs, such as taxol or vinca, that directly interact with microtubules and their expression level can be associated to the type of prognosis. Moreover, the use of drugs targeting MAPs can be less toxic than drugs directly interacting with microtubules.

We recently demonstrated the ability of a 2-benzodiazepine-3-one derivative (1-(4-amino-3,5-dimethylphenyl)-3,5-dihydro-7,8-ethylenedioxy-4h2,3-benzodiazepin-4-one, hereafter named 1g) to arrest the cell cycle progression in human leukemia Jurkat T cells and HeLa cells by affecting microtubule dynamics and altering the mitotic spindle formation with the formation of multipolar spindles without centrosome amplification [40,41]. In the case of HeLa cells, the molecule does not appear to affect microtubule organization in the interphase and it does not seem, based on in-vitro polymerization analysis, to interact directly with microtubules but rather through some MAPs. This behavior opens up the possibility of a specificity of the interaction of this molecule for different cell types, due to the presence of different MAPs in different cell lines. Since the new molecule is supposed to be able to cross the blood–brain barrier [42], it could represent an interesting candidate in the fight against brain tumors.

In this work, we studied how 1g interferes with the replication, migration, and mechanical properties of the U87MG cell line. We found that 1g, in the case of glioma cells, not only affects cell replication but it also strongly affects migration capability. We suggest that 1g, at the tested concentrations, acts by altering microtubule dynamics without inducing depolymerization. In terms of cellular morphology and migration capacity, by producing a loss of cell polarity and impeding migration, this molecule proves to be more effective in the case of cells that rely significantly on microtubules for these functions, such as glioma cells. We also found pieces of evidence suggesting an interaction of the molecule at issue with the microtubule/plasma membrane attachment sites causing both a loss of polarity and an aberrant mitotic fuse formation due to the disorganization of the astral microtubules.

## 2. Results

### 2.1. 1g Reduces U87MG Spheroid Matrix Invasion

Multicellular spheroids are an advantageous model system for studying cell invasion in a 3D environment and the role of the mechanical properties of the cell environment on cell invasion [43,44,45,46,47]. Recently, they attracted much interest because they ensure conditions such as the presence of an oxygen gradient from the peripheral zone to the central one of the aggregate, with the internal area in some cases reaching a necrotic condition, and they include the presence of intercellular interactions, and the presence of the extracellular matrix, making the cell environment more similar to that of in-vivo cancer tissues. We first measured the expansion rate of U87MG spheroids embedded in a basement membrane matrix (Matrigel^®^, Corning Inc., New York, NY, USA, 10 mg/mL protein content). Figure 1a shows optical microscopy images of the U87MG spheroid expansion under control conditions and exposed to a 20 μM 1g concentration, 35 h and 68 h after their embedding. Control spheroids showed a larger area increase in comparison to those treated with 1g, at both time points. We then analyzed the projected area of the spheroid at the equatorial plane, normalized with respect to the initial area, as a function of time (Figure 1b). The reduced expansion of the U87MG spheroids establishes that 1g can affect cell behavior in the 3D configuration, where cells can experience strong cell/cell interactions. From the area expansion shown in Figure 1b, it is, in principle, possible to calculate the volume increase, assuming the complete shape of the structure as spherical. Interestingly, the central region of the control spheroids shows faded edges, indicating a prevalent migratory behavior of cells, whereas the central region of spheroids exposed to 1g shows a well-defined and persistent edge, indicative of a reduced cell migration from this region (see Appendix A).

Finally, we analyzed the migration mode of the expanding U87MG cells by time-lapse microscopy (see Appendix A). Whereas the migration in the control condition appeared mainly as a collective phenomenon, with cells extending long processes ahead, 1g-treated cells, in addition to the limited expansion, moved more individually, remaining mainly round-shaped and suggesting a more amoeboid-like migration mechanism (Figure 1c and Appendix A). These experiments indicate that 1g is able to reduce U87MG spheroid invasion in a Matrigel matrix.

### 2.2. 1g Reduces the Contractility of U87MG Spheroids Embedded in Matrigel

To extend the results obtained with spheroid invasion experiments, we concentrated on traction force microscopy (TFM) experiments on spheroids embedded in the same basement membrane matrix. In this context, we based our analysis on the work by Mark et al. [48]. We thus measured the contractile properties of cell aggregates as a function of time in the presence or absence of 1g. Contractility is an important parameter in analyzing the invasive behavior of cells in a 3D environment, as cells exploit the traction force applied on the fibers of the matrix to migrate [49,50]. The specificity of the present approach is the measurement of the traction forces/contractility for a cell aggregate and the fact that the analysis is applied in the context of the non-linear elastic behavior of the embedding matrix (the matrix effective Young modulus changes, specifically increasing upon an increase of the deformation). In this way, the physiological matrix environment of a tumor sphere is better represented with respect to synthetic matrices which are typically characterized by a linear elastic regime. To perform this analysis, we measured the displacement of micron-sized fiducial markers represented by 5 μm diameter latex beads at the equatorial plane of the spheroid, assuming a spherical symmetry for the spheroid and its environment and reproducing the experiment in silico after establishing a lookup table for the specific matrix we used [51]. The assumption of spherical symmetry of the system greatly simplifies the requirements for reconstructing the force field. In our case, the matrix we exploited was composed of Matrigel^®^, which has an almost linear elastic behavior. Anyway, in the contractility analysis, we nonetheless considered the presence of a small non-linear behavior [52]. Figure 2a shows three different frames, corresponding to different time points of the experiment, of U87MG spheroids under control conditions and in the presence of 20 μM 1g. Figure 2b reports the situation at the end of the experiment, highlighting the displacement vectors for the beads. Figure 2c reports the mean contractility of the spheroids as a function of time. The spheroids produced a monotonic contraction of the gel over time with an almost radial symmetry. Appendix A show the evolution of the experiment over 46 h of imaging with frames acquired every hour for both a control spheroid and a spheroid in the presence of 20 μM 1g. It is important to note that we never observed a pushing process of the expanding spheroids on the beads. The presence of this behavior could eventually be related to the duplication of cells not compensated by the invasion and a consequent expansion of the spheroid volume or, even if the spatial resolution in our experiment is not sufficient to measure this effect, to an amoeboid cell migration mode. In our case, the traction force of invading cells probably overcomes the cell duplication process which, in the time interval we considered, is however limited. In this case, contractility is represented by (pressure) x (surface area) and it is expressed with units of force. We found that the presence of 1g decreases, from the beginning of the experiment, the contractility produced by the cells, till a stabilization phase occurs. The value we obtained for the contractility after 12 h (about 150 μN) is compatible with the value reported by Mark et al. on similar cells [48] and it is important to emphasize that the obtained force value points to the relevance of the collective behavior of the cells as different with respect to the simple sum of individual behaviors, especially over long time intervals. This could be due to the larger force applied by the spheroid and the strain stiffening effect of the extracellular matrix, even in the case of an almost linear behavior of Matrigel. From a physiological point of view, the strain stiffening phenomenon and the possible alignment of fibers of the matrix could lead to an increased cell invasion. These experiments indicate that, in the presence of 1g, the limited spheroid invasion could be related to a decreased contractility of the cells.

### 2.3. 1g Decreases the Surface Tension of U87MG Spheroids

From a physical point of view, it has recently been highlighted how typical physical properties of cell aggregates share some similarities with the behavior of solids and liquids, and the relevant mechanical parameters are defined by their stiffness (Young modulus), viscosity, and surface tension [43,53,54]. In particular, the effective surface tension in liquid drops, such as in spheroids, controls both the size of the corresponding spherical shape and the possibility of the detachment of smaller droplets from the main structure, a process that resembles the initial step of the metastatic pathway. Accordingly, it has been found that spheroids composed of high metastatic potential cells are typically characterized by a lower surface tension compared to non-metastatic cells [55]. Different techniques can be exploited to measure the passive mechanical properties of spheroids such as their Young modulus, viscosity, and surface tension [53]. Here, we exploited the Micropipette Aspiration Technique (MAT) [56,57]. Figure 3a shows a bright field image of an U87MG spheroid at the beginning of an aspiration experiment inside a glass micropipette. Figure 3b reports the evolution of the spheroid projections inside the micropipette as a function of time when a pressure step of 20 cm H_2_O (1960 Pa) is initially applied, and released after 40 min. Two curves, representative of the control and 1g-treated spheroids, are reported. The behavior immediately after the pressure step is related to the elastic properties of the cell/ECM aggregate. This phase is followed by a viscous flow of the aggregate inside the micropipette. Different mechanical phenomenological models can be exploited to describe the relaxation behavior as a function of time and, in our case, we used the modified Kelvin–Voigt model depicted in Figure 3c [58]. The specific behavior for the creep response of the spheroid can be written as follows:(1)$$L\left(t\right)=\frac{F}{k_{1}}\left(1-\frac{k_{2}}{k_{1}+k_{2}}e^{-\frac{k_{1}k_{2}}{\eta_{2}\left(k_{1}+k_{2}\right)}t}\right)+\frac{F}{\eta_{1}}t$$
where *L*(*t*) is the spheroid projection inside the micropipette and *F* is the applied stress, which in this case is given by the following expression:$$F=∆P-∆P_{c}$$
where Δ*P* is the effective pressure step that is applied and it represents the parameter we can control, and Δ*P_c_* is the critical pressure, i.e., the pressure above which the spheroid starts to move inside the micropipette.

Δ*P_c_* can be related to the Laplace law for the aspirated spheroid with a surface tension. In fact, we can write the following:$$∆P_{c}=2\gamma \left(\frac{1}{R_{p}}-\frac{1}{R}\right)$$
where *R_p_* is the radius of the micropipette and *R* is the radius of the spheroid. Experimentally, Δ*P_c_* could be measured by considering the minimum applied pressure difference that initiates the continuous flow of the spheroid tongue inside the pipette. In practice, this is the Laplace pressure due to the curvature imposed by the pipette size.

We preferred to evaluate $∆P_{c}$ following a procedure based on the flow speeds of the spheroid inside the capillary, both inward and backward (see Section 4: Materials and Methods). Once Δ*P_c_* has been evaluated, the corresponding surface tension of the spheroid can be obtained. According to this procedure, we obtained, in the case of control spheroids, $∆P_{c}=\left(11.6\pm 2.3\right)\;cmH_{2}O\left(1.137\;kPa\right)$ and $\gamma =\left(19.6\pm 1.0\right)\frac{mN}{m}$ (*n* = 3). In the case of spheroids exposed to a 20 μM concentration of 1g we have the following: $∆P_{c}=\left(7.8\pm 3.0\right)\;cmH_{2}O(765\;Pa)$ and $\gamma =\left(13.2\pm 2.0\right)mN/m\;$(*n* = 3).

We analyzed three spheroids for both conditions and the observed trend was always the same, corresponding to a decrease in surface tension when spheroids have been exposed to 1g for 24 h (see Appendix A). In principle, a decrease in surface tension should favor the detachment of single cells from the aggregate, enhancing the possibility of metastasis [55]. However, here we think that the reduced tissue surface tension is due to the increase of the cortical tension of U87MG cells after 24 h of exposure to 1g, as we will describe below. The increased cortical tension decreases the cell–cell contact area and adhesion, reducing the overall tissue surface tension.

A linear fit to the last part of the aspiration data provides both the speed of the entering spheroid (*v_in_*) and the intercept for *t* = 0 that corresponds to $\delta =\frac{F}{k_{1}}$ (see Section 4: Materials and Methods). In the two different cases we obtain the following: $\delta_{control}=\left(55\pm 10\right)\mu m$ and $\delta_{1g}=\left(87\pm 12\right)\mu m$ (Appendix A). At the same time, we can evaluate the Young modulus coming into play for times longer than the characteristic one defined by the exponential dependence. For the control spheroids (see Section 4: Materials and Methods) we obtain the following: $E_{1,control}=\left(449\pm 40\right)\;Pa\;$whereas, for spheroids exposed to 1g, we have $E_{1,1g}=\left(412\pm 40\right)\;Pa$. Considering Ref. [59] and the fact that we treated the inner surface of the glass pipette to avoid adhesion as much as possible, we can calculate the viscosity of the aggregates (see Section 4: Materials and Methods). In the case of the control spheroids we have $\eta_{control}=\left(2.6\pm 0.4\right)\times \frac{10^{5}Pa}{s}\;$ and for the spheroids exposed to 1g we have $\eta_{1g}=\left(1.9\pm 0.3\right)\times 10^{5}Pa/s$.

### 2.4. 1g Reduces Cell Proliferation in 2D Cell Cultures

Data obtained with U87MG spheroids established that 1g can limit the traction force exerted by these cells in a 3D environment and, accordingly, decrease their invasion capability. At the same time, we observed a decrease of the surface tension upon the exposure of the cells to 1g, with cells behaving in a more individual manner and limiting the traction force they can apply to eventually orient fibers of the ECM. In addition, previous experiments using different cell lines showed that 1g arrested the cell cycle in the G2/M phase [60]. We then moved to 2D cell cultures and we found that 1g is able to inhibit cell viability in a dose- and time-dependent manner (Appendix A). Figure 4 shows examples of the effects of a 20 µM 1g delivery on the morphology of U87MG cells within a very short time interval (~15 min). The cells rapidly lose their polarization with a fast retraction of the processes.

Co-cultures experiments with different cell lines, such as NIH-3T3 cells, showed that the phenotypic changes we observed were cell-type specific (Appendix A). The presence of 1g, at the concentration that produced the retraction of processes in U87MG cells, had negligible effects on the morphology of NIH-3T3 cells.

We also analyzed the seeding of U87MG cells in the presence of 1g. Observing by time-lapse optical microscopy the behavior of the cells from the moment they are seeded, we found that U87MG cells, in the presence of 1g, are able to adhere and spread on the substrate but they are not able to polarize, in contrast to what happens in the absence of 1g (Appendix A). In fact, in the presence of the drug, cells flatten as round objects on the surface and they are characterized by a strong dynamic behavior along their perimeter, continuously trying to break the symmetry, but without success.

In addition, we measured the effects of 1g on U87MG cell adhesion (Figure 4b). The cells were treated with the compound at 5–50 µM for 30 min and with 1g 20 µM for 6 h (pre-treated cells). In the latter case, the cells were washed out after the incubation time and plated with the untreated medium. The 30-min adhesion capability of U87MG cells, to the ECM monolayer, is increased in the presence of different concentrations of 1g. In particular, the compound significantly increased the adhesion of the cells at the highest concentration tested. Interestingly, when we performed a 6 h pre-treatment with 1g 20 µM, the cells adhered much better than the cells treated for 30 min at the same concentration. Indeed, the absorbance reached the value obtained at a higher concentration such as 50 µM. These data support and confirm the previous finding, indicating that the cells react to the 1g treatment, as shown by peripheral dynamism, but without success.

To investigate the reversibility of the 1g treatment in the context of its effect on cell polarity, we exposed U87MG cells to a 20 µM 1g concentration for 5 h, and we then washed out the drug and returned it to the normal culture medium. Appendix A show that, after the washout of 1g, cells re-acquire their normal polarization and start migrating again, reaching a normal mitotic process.

Considering cell morphology, the observed behavior is similar to what Venere et al. found in glioblastoma multiforme cells, where the molecular motor Kif11 was inhibited by ispinesib [61]. The interesting aspect of molecules targeting this specific molecular motor is related to the fact that they would work against the formation of the mitotic spindle without affecting the transport of vesicles along microtubules in interphase which is fundamental in the case of neural cells. Moreover, Kif11 has been found to be relevant also in cell migration [62] and its inhibition would affect both the proliferative and invasive capability of cells. Therefore, we analyzed whether the effect of 1g could be related to the inhibition of Kif11. However, the inhibition of Kif11 would block the centrosome separation, leading to the formation of a so-called monopolar spindle (monoaster). Both in U87MG cells and in the previously studied HeLa cells [40], the presence of separated asters identified by the presence of γ-tubulin, was observed, as shown in Appendix A, aiming against a Kif11 role. Moreover, the distance among the different γ-tubulin-marked asters in 1g-treated cells is lower than in control cells, pointing to an effect of 1g on the dynamics of microtubules. It is interesting to consider that 1g was able to induce multipolar spindles even in the absence of centrosome amplification for HeLa cells, which is also reported in the case of paclitaxel at clinically relevant concentrations [63].

We already demonstrated that 1g is able to block U87MG cells in mitosis [60]. Here, by exploiting time-lapse imaging of U87MG cells exposed to 1g, we found that, after rounding and spreading on the surface, the cells can get spherical to enter mitosis and remain in this situation for a long period of time. Figure 5a shows that, compared to an average time of 20 min spent in the mitotic phase for untreated cells, cells treated with different 1g concentrations retained the spherical shape for a period of time strongly correlated with the 1g concentration. During this time, the cells continuously try to exit mitosis, as shown in Appendix A. By using Sir-tub as a live fluorescent marker for tubulin, we followed the tubulin reorganization during the mitotic process of U87MG cells exposed to 20 µM 1g (Appendix A). It appears that the mitotic fuse is immediately disorganized with the formation of multipolar spindles. It is relevant to observe that even cells that had already started mitosis when exposed to 1g, rapidly disorganized their mitotic fuse, suggesting a mechanism based on the alteration of the tension on microtubules affecting the stabilization of the mitotic spindle.

When cells are in the rounded shape, presumably in mitosis, we observed different exit strategies (see also below): (1) cells die during the mitotic phase; (2) cells flatten again on the surface without dividing; and (3) by undergoing an aberrant mitosis, the cells initially appear to produce more than two daughter cells but, after this initial phase, they once again flatten on the surface as a single cell or eventually produce more than two daughter cells with a multipolar division and producing aneuploid progeny (Figure 5b). In the last two cases we can find, inside the cell, multi-mininuclei defined by lamin staining but without genetic material inside (see Appendix A). In the latter case, the cell is supposed to divide its duplicated chromosomes in multiple directions. Immunofluorescence analysis shows that in many cases, cells exposed to 1g and blocked in mitosis, are characterized by a multipolar spindle as shown in Figure 5c. This behavior resembles the typical effect of microtubule stabilizing agents such as paclitaxel during the mitotic process, suggesting that the delay in mitosis for 1g is related to the Spindle Assembly Checkpoint (SAC) [64].

Figure 5d shows an example of a time-lapse optical microscopy analysis showing what happens to cells after a prolonged exposure to 20 µM 1g. From the analysis performed by prolonged time-lapse optical microscopy, we obtained that the majority of the cells are able to perform a mitotic slippage, whereas a small number of cells die trying to perform the first mitotic process. The experiment we performed was limited to 50 h of time-lapse imaging, and only a very small number of cells that had performed the mitotic slippage were observed to die during the following interphase stage. During the entire time of the time-lapse analysis, we never observed a cell undergoing a new attempt of mitosis after the mitotic slippage. We have to consider anyway that, given the typical duplication time for control U87MG cells of 20 h, the possibility of a second mitotic attempt is quite low. At the same time, it appears that 1g strongly affects the time spent in mitosis but is not able to kill a cell directly in mitosis. Cells preferentially exit mitosis and, without a passage of the cells through mitosis, 1g is not able to kill cells.

### 2.5. 1g Inhibits the Migration of GBM in 2D Cultures

In addition to the effect on U87MG cell duplication, we found an effect of 1g on cell migration specific for U87MG. GBM cells are endowed with a very high infiltration ability exploiting physical structures in the brain environment such as blood vessels and axons, and the possibility of affecting both duplication and migration would add another potential use of 1g to fight against cancer. We have previously shown that cells incubated with 1g changed their shape towards a round and non-polarized geometry already after 1h of incubation time, suggesting a possible role of the compound on cell motility. To examine the effects of 1g on collective GBM migration activity in 2D, wound healing assays were performed. As shown in Figure 6, the compound was able to inhibit the migration of U87MG cells. Indeed, cells incubated with 1g migrated within 24 h across a smaller area than control cells incubated with vehicle medium (0.1% DMSO). The recovered wound area was almost 63% in the control cells, and 37% in the cells treated with 20 μM 1g.

The molecular basis for the glioma cells migration has been largely investigated and it has been found that, at least in 2D, their migration is not affected by drugs that can block myosin, pointing to a migration pattern more similar to fibroblasts and highlighting the specific relevance of microtubules in this essential cellular process [22]. In the case of constrained 3D migration, the role of the myosin molecular motor is instead fundamental, pointing to the difference between 2D and 3D models. Here, we have focused on 2D migration to also obtain mechanistic details about the 1g mechanism of action. It should be emphasized that the GBM cell migration process is sensitive to the mechanical properties of the substrate on which cells reside [16,65,66]. At the same time, it has been shown that the effect of drugs on the same type of cells depends on their mechanical environment [67,68] and drugs affecting molecular motors or cell–substrate adhesion complexes could have different effects on cell migration properties depending on the stiffness of the substrate [69]. In order to evaluate the role of the mechanical properties of the substrate and the effect of 1g on the 2D cell migration properties of isolated/non-interacting U87MG cells, we seeded the cells on PDMS substrates with different Young moduli and on plastic. The range of substrate stiffness we considered spans the interval from 0.2 kPa to 300 kPa for PDMS supports, whereas, for the plastic substrate, we can assume a Young modulus of a few GPa. In each experiment, control (DMSO) cells were compared with cells exposed to different 1g concentrations. As expected, cells on softer substrates had a smaller adhesion area in comparison to cells adhering to more rigid supports (see Appendix A). We then analyzed the migration of individual cells on the different substrates exploiting the Mean Squared Displacement (MSD) parameter as a function of time.

Figure 7 reports different aspects of the migration analysis. We compared the MSD of the cells as a function of time for different substrate rigidities in Log–Log plots. To visually compare the slope of the curves, we also inserted the reference for the ballistic (MSD∝t^α^, α = 2) and the random walk Brownian migration (MSD∝t^α^, α = 1) to visually compare the slope of the curves. In the inset of each plot in Figure 7, we also report the values of the MSD at the end of the experiments, considering the different Young moduli of the substrates. Control (DMSO-treated) cells display mechano-sensitivity in their cell migration as their final MSD is higher for substrates of intermediate stiffness, with a maximum at about 32 kPa. The limited resolution relating to the different values of substrate rigidity does not allow the value that assures the maximum explored area to be identified with sufficient accuracy, but we can anyway state that the explored area has a biphasic behavior. In Appendix A, the behavior of the MSD value for the different 1g concentrations for each substrate stiffness is reported. From the plots of the MSD values as a function of time and 1g concentration, the presence of different migration behaviors induced by 1g is not immediately evident and it is not clear whether these changes are eventually affected by the rigidity of the substrate. On the other hand, considering the plots of the derivatives of the MSD as a function of the logarithm of time, providing the corresponding exponent (Appendix A), we can observe that the presence of 1g, starting from concentrations of 10 μM, favors the transition to a behavior more similar to a pure random walk process. This is likely a consequence of the loss of cell polarity. We can speculate that each time a cell tries to establish a polarization, biochemical signals stop this process and the cell tries a new polarization in another direction producing a random walk behavior. This change in the behavior is much more evident on rigid substrates such as plastic or PDMS 1:25 with a Young modulus of about 300 kPa. In the case of soft substrates, the value of the exponent (Appendix A) suggests that the migration is more of a purely random process and 1g does not appreciably change this behavior. Interesting differences in the behavior could be found considering the autocorrelation direction of the cell migration. In this case we considered the autocorrelation of the cosine function in which we measured the angle between two consecutive steps of migrating cells. The faster the decay of the autocorrelation function, the faster is the loss of memory of the cell migration direction. In Appendix A, the direction autocorrelation trends as a function of the 1g concentration, grouped according to the substrate rigidity, have been reported. Interestingly, for more rigid substrates such as plastic or PDMS 1:25, the decay of the direction autocorrelation function is slow over time and the presence of 1g seems to moderately accelerate the decrease. This aspect could be related to the increased adhesion area of the cells on rigid substrates and to the increased complexity of the cytoskeletal structure. As a consequence, changing the cytoskeletal structure to modify the migration direction requires a longer period of time, maintaining a high value of the direction autocorrelation function for a longer time interval. For softer substrates the decrease is fast also in the absence of 1g and 1g is only active in further decreasing the autocorrelation function for the softest conditions of the substrate (Appendix A). Also interestingly, for very soft substrates, we observe a negative value of the autocorrelation function. This behavior could be interpreted as consecutive movements of cells in opposite directions. It is likely that the consecutive attempts of cell polarization occur from positions along the cell periphery separated by about 180°.

We next wanted to compare the effect of 1g on the migration behavior of cells for which 1g has little effect on the morphology. We thus considered how the migration of NIH-3T3 cells are affected by different concentrations of 1g. Appendix A shows that while at concentrations up to 10 μM 1g has negligible effects on NIH-3T3 cells, compared to control cells, at 20 μM we observed a substantial decrease in cell migration (measured in terms of MSD and the corresponding diffusion coefficient).

Interestingly, we typically found the development of actin stress fibers after 24 h of the incubation of U87MG cells with 1g, as shown in Appendix A, where a typical immunofluorescence image of U87MG cells is compared with cells exposed to nocodazole, which is known to produce stress fibers as a consequence of microtubule destabilization. In a previous investigation [40], we found that 1g seems to have an effect on microtubules in cells. Indeed, 1g did not affect microtubule cold depolymerization, excluding a possible stabilizing effect for 1g, but, at the same time, 1g slowed down the regrowth kinetics of depolymerized microtubules [40]. In Hela cells, experiments showed that 1g affects the growth rate of microtubules, decreasing it, but not the growth life-time [40]. Interestingly, the effect of 1g does not seem to act directly on microtubules, because self-assembling turbidity experiments in-vitro show no difference when 1g is added. Accordingly, 1g most likely acts on some MAPs at variance with what happens with other molecules such as taxans or vinblastine which are known to have a direct binding site on microtubules. We also expect that this effect will have consequences on actin organization. Apart from microtubule depolymerization, also the post-translational modification (PTM) of microtubules is known to affect the actin structure, for example by inducing the release of the guanine nucleotide exchange factor GEF-H1 and, as a consequence, the activation of RhoA and the formation of an enhanced stress fibers complex [70]. The effect of 1g on the actin structure was also tested on G166 cells. In this case, immediately after the injection of 1g, we observed by immunofluorescence a strong reinforcement of the cortical actin layer (Appendix A). In these types of glioblastoma cells, there is a stronger cell–cell interaction and cells do not retract upon interaction with 1g. In this way, it is possible to highlight the effect of 1g on the cell cortical layer. In fact, in the case of U87MG cells, the rapid retraction of the cell processes and the rounding of cells make it difficult to distinguish between an F-actin reinforcement and an increase in the immunofluorescence signal due to a volume reduction of cells.

### 2.6. 1g Increases the Cell Stiffness of GBM Cells as Measured by Atomic Force Microscopy

In addition to the mechanical properties of multicellular aggregates, we also investigated the mechanical properties of single cells exposed to 1g. It has been reported that the mechanical phenotype of cancer cells may be different from that of healthy cells. For example, more deformable cells could be endowed with an advantage in the confined migration through narrow pores of the extracellular matrix [71,72]. As a consequence, the metastatic potential of tumor cells has been found to correlate with their mechanical properties, typically with softer cells exhibiting a higher metastatic potential [73,74]. It appeared therefore interesting to evaluate the effect of a potential drug on cell mechanical properties at the single cell level [75] and, specifically, on the possibility of increasing cell stiffness to decrease its metastatic ability. We used atomic force microscopy (AFM) to compare the stiffness of U87MG cells treated for 24 h with 20 μM 1g with untreated cells. Based on previous time-lapse imaging experiments, after 24 h, cells that can be analyzed for their mechanical properties are typically cells that tried to enter mitosis but failed and turned back to interphase in a flattened morphology. Figure 8 shows the results of the single cell mechanical properties investigation. We initially measured the Young modulus by performing force curves with a fixed vertical speed of the cantilever base and analyzed the curves with a modified Hertz model (see Section 4: Materials and Methods) to obtain the mechanical parameters of interest. Figure 8d shows the distribution of the Young moduli that we obtained on untreated U87MG cells or on a U87MG cell population maintained in a 20 μM 1g solution for 24 h. Both distributions follow a Log-normal behavior and cells treated with 20 μM 1g for 24 h appear more rigid than control cells. All the curves were obtained using a spherical colloidal probe with a diameter of 5 μm and positioning the tip in a region of the cell near to the nucleus. However, the use of the Young modulus as a marker for cell metastatic potential has been questioned. In some cases, it has been found that cells with a higher metastatic character are stiffer than healthy cells of the same type [76]. The appearance of a metastatic character of cells is closely related to the structure of the cytoskeleton and a close relationship between cancer and cell mechanics is to be expected. Not only the elastic behavior is relevant for cell functions, but also the viscous (or time dependent) aspects can be relevant and these properties can be evaluated by a dynamic analysis as a function of the frequency of the stimulus [77,78,79]. To account for these possible contributions, we performed a dynamic mechanical analysis by AFM on U87MG cells untreated and treated with 1g. We explored a range of frequencies from 30 Hz to 400 Hz, and obtained, as shown in Figure 8e, a slowly and linearly (in a Log–Log plot) increasing value of the elastic and viscous components of the complex shear modulus for control cells. This trend points to a power law dependence, a typical behavior already observed on other living cells [78]. In Figure 8f, the comparison of the complex shear modulus as a function of frequency between control cells, cells treated for 24 h with 20 μM 1g, and cells treated with 20 μM nocodazole for 24 h is reported. A comparison with the drug Nocodazole was considered here because of its well-known effect on microtubules depolymerization and the consequent enhancement of stress fibers structures [80,81] and because the organization of the stress fibers of 1g-treated U87MG cells after 24 h resembles that of the same cells exposed to nocodazole (see Appendix A). It has been suggested that, upon an increase of the probing frequency, the viscous contribution may become stronger than the elastic one, defining a sort of phase transition. The relative contribution of the two components is expressed by the loss tangent parameter, defined as the ratio of the viscous to the elastic parameter (*η* = *G*″/*G*′). The crossover defines the phase transition and the value of this parameter has been found in the range 100 Hz–1000 Hz [78,82]. In our case, we did not observe any crossover in the range of frequencies we explored, both for control cells and for cells treated with 1g or nocodazole. We found that both components, the elastic and the viscous ones, were increased by the 1g treatment both at low and high frequencies. We also observed an increase of the slope of the curve representing the shear moduli, G′ and G″ (and of the absolute shear modulus—Figure 8f), as a function of the probing frequency for treated cells. This behavior points to cells that are more similar to a fluid system with respect to the control cells. In the literature, the use of drugs that stabilize the actin cytoskeleton, such as caniculin, typically reduces the slope of the curve representing the shear modulus as a function of the probing frequency. In our case, we used a spherical tip and the indentation produced in the cell (around 50 nm) is such that it probes mainly the region close to the cortical actin and the possible detachment of the cell plasma membrane from the cortical layer could be responsible for this behavior. Anyway, to further confirm the results obtained by dynamic mechanical analysis with AFM, we performed an analysis of force curves on U87MG cells, untreated and treated with 1g, according to the Ting model [83]. Using this type of analysis, from a single force curve, considering both the approaching and retracting portions, it is possible to obtain both the elastic and viscous components of the cell mechanical properties (see Section 4: Materials and Methods and Appendix A). Even from this type of analysis, it results that 1g favors an increase of both the elastic and the viscous moduli (see Appendix A).

### 2.7. The Modulation of the Principal Pathways Involved in Cell Adhesion and Migration

The first response of U87MG cells to the presence of 1g is their loss of the polarization, suggesting that a disassembly of Focal Adhesion (FA) complexes and cortical actin could be involved. Accordingly, we investigated the behavior of actin and ERM molecules, which are responsible for the membrane cortical cytoskeleton interactions, both for actin and microtubules and have a strong influence on the cortical structure. From both immunoblotting and immunostaining, we find a strong increase of the pERM/ERM ratio upon the addition of 1g into the culturing medium (Figure 9a). The phosphorylated form of the ERM proteins corresponds to the stabilized state of the contact between the membrane and the cortical cytoskeleton. It is interesting to note that this phenomenon occurs immediately after the injection, highlighting a reaction of cells to a perturbing event. It is possible that, at longer incubation periods, the increase of the pERM/ERM ratio is due to the arrest of cells in mitosis, when pERMs are intrinsically increased. Figure 9b shows that pERM colocalizes with actin filaments with the formation of actin rings. It is important to note that the iper-phosphorylation of ERM proteins within minutes after a 1g injection has been observed for cell phenotypes other than glioma cells (for example for NIH-3T3). However, only in the case of U87MG, we observed a strong effect on cell morphology and neurite retraction. This finding is consistent with the results obtained by Areti A et al. [84], where a partnership between ERM and Rac-1 played an important role in neurite outgrowth.

The retraction of the processes within minutes of the addition of 1g could also be explained by an increased phosphorylation of the myosin II light chain (MLC) and, as a consequence, an increase in the traction force exerted by the cell [85]. To analyze this possibility, we considered the effect of 1g in the presence of blebbistatin, which is known to block the activity of the myosin II head by inhibiting myosin ATPase activity and, consequently, the traction force due to the actin–myosin complex. The presence of 20 μM of 1g and 20 μM of blebbistatin induced a retraction of the processes with a reduced kinetics, showing that the increased traction force could not be the main aspect responsible for processes retraction (see Appendix A). At the same time, as already reported in the literature [86], blebbistatin alone does not affect the U87MG migration while producing an increase of the number of processes. In the presence of both 1g and blebbistatin, we again observed an arrest of the cell in the mitotic phase. It is to be considered that blebbistatin has also been reported to affect the formation of the mitotic spindle [87] and, accordingly, we cannot draw conclusions about what happens to the spindle structure since the two drugs may have the same effect. We also tested the Rho-kinase inhibitor Y-27632 together with 1g. Once again, Y-27632 alone is not able to produce a notable effect on U87MG cell migration. In the case of Y-27632 together with 1g, we observed a relatively smaller retraction of protrusion extension with respect to 1g alone (see Appendix A) and cells, after having attempted mitosis for a long period of time, in almost all cases exited this phase and returned to interphase with the complete absence of polarization. It is to be considered, however, that myosin II is relevant in the cytokinesis process and a complete mitotic process could be impeded by this effect. Additionally, it has also been reported that molecules interacting with microtubules, as we hypothesized for 1g, can affect Rho and Rac1 GTPase signaling and consequently cell contractility and migration [88]. Moreover, Areti et al. [84] showed a mutual cooperation of Rac-1 GTPase and ERM protein in neurite outgrowth. However, in our experiments, we observed a decrease of Rac-1 expression after 12 h of 1g treatment (Figure 10). This is in contrast to what happens with high concentrations of nocodazole, which is known to depolymerize microtubules releasing guanine-nucleotide-exchange-factor (GEF-H1), which is an upstream activator of Rac-1, and increasing cell contractility with the development of stress fibers. Once again, these data suggest that 1g acts differently compared to nocodazole. The still-present retraction of processes, albeit with a reduced dynamics, in the co-presence of 1g with blebbistatin or Y-27632, suggests that the activation of myosin II is not responsible for this change in the shape of U87MG cells.

The loss of cell polarity and a strong decrease of the explored area by cells stimulated the investigation by the immunoblotting of other pathways involved both in cell migration and cytoskeletal reorganization. We then evaluated the ability of 1g 20 μM to modulate the expression of structural proteins α-Actinin, Paxilin, Vinculin, and focal adhesion kinase (FAK), which, among others, are responsible for cell migration and polarization. As reported in Figure 10, 1g did not modulate the expression of both the structural protein α-Actinin and Paxilin at any of the time points tested. On the other hand, Vinculin, which is involved in focal adhesions, showed a significant decrease in protein immuno-density already at 5 h which remains downregulated up to 12 h of 1g treatment compared to the control. The data obtained with FAK protein expression showed that the band intensity did not change all over the period of treatment. Interestingly, the results on the protein expression of LIMK and cofilin/p-cofilin, showed the ability of 1g to increase the level of the LIMK protein after 5 h of the incubation time, but the band intensity returned to the level of the control after 12 h of the 1g treatment, whereas the phosphorylated form of cofilin increased its expression already after 5 h of incubation and lasted for the entire time tested (Figure 10). This last result is interesting in terms of tumor growth and progression, since cofilin, whose phosphorylated form is the inactive one, has been proposed as a possible booster of glioma progression [89]. The results obtained for the LIMK and p-cofilin proteins suggest a stabilization of the actin structure. Taken together, these findings suggest that 1g, through its action on microtubules, is able to affect actin organization following a different pathway compared to other molecules, such as nocodazole, which act directly on microtubules.

## 3. Discussion

Glioblastoma multiforme (GBM) is the most common primary tumor of the central nervous system. Its very poor prognosis is mainly related to its extremely high invasive potential, which allows its cells to migrate along vessels and through the white matter in the brain, probably exploiting specific mechanisms for different environments. Therefore, it is urgent to develop new compounds for GBM treatment that can be used alone or in combination with existing drugs to enhance efficacy and minimize side effects. Extensive research has focused on the cytoskeletal features of tumor cells to evaluate the dynamics of cell motility and invasion. Considering this, we conducted a detailed study on how a new benzodiazepine derivative (1g) affects cellular mechanics, proliferation, invasion, and migration of the glioblastoma cell line U87MG.

We conducted invasion experiments by embedding spheroids in a Matrigel environment. Spheroid expansion experiments did not allow us to fully distinguish between the contributions of cell invasion and cell duplication. This distinction is particularly important for U87MG cells exposed to 1g, as biochemical and optical microscopy analyses of 2D cell populations indicate that 1g induces an arrest of U87MG cells in the G2/M phase, functioning similarly to a cytostatic drug. However, in our case, we can assume that the most significant factor influencing the observed expansion of U87MG spheroids is cell diffusion, which relates to random migration along with a form of directed motility away from the core region. This directed motility is likely attributable to a gradient in nutrients [90], although the exact process is not completely understood. This assumption is supported by taking into account that only a limited time (65 h) is considered, suggesting that proliferation is not the primary factor governing the expansion behavior.

To better distinguish between the processes of proliferation and invasion, it would be useful to extract quantitative information from the process reported in Figure 1. The growth of spheroids has been described using various mathematical models, with the Gompertz [90,91] model being the most widely employed. This model consists of several phases, including a latent period, an optimal growth phase, and a plateau phase. The mathematical representation of spheroid growth is characterized by a sigmoidal behavior in the spheroid radius as a function of time. The initial latent period corresponds to the formation of a compact cell aggregate, followed by an exponential growth phase. This is followed by a linear growth phase, during which the rim of proliferating cells remains constant, and, finally, a saturation phase is reached where growth is balanced by necrosis. Although the growth trends we observed align with this model, it is essential to consider the invasive nature of glioma spheroids when they are embedded in a 3D gel. For smaller spheroids, such as those examined in this study, it is crucial to distinguish the two different regions: the core and the rim. These regions exhibit different behaviors regarding proliferation and dispersion. The core region of spheroids is typically characterized by rapidly proliferating cells with low diffusion properties. In contrast, the rim consists of cells that exhibit a higher diffusion behavior but lower replicative properties, in accordance with the go-or-grow hypothesis. Mathematical models have been developed to simultaneously describe cell duplication and invasion in spheroids. These models commonly utilize various parameters to characterize the invasion and growth of spheroids, including a diffusion constant (D), a maximum cell density to account for growth inhibition at very high densities, a coefficient representing the number of cells moving away from the core’s boundary per unit time, a factor for cell duplication time, and a directed velocity—likely due to the gradients of nutrients or signals from metabolic byproducts of the spheroid cells. Using these models could help evaluate whether the exposure to 1g, for example, favors an expansion in which diffusion dominates over directed motion. However, fitting these models to experimental data requires a clear distinction between the core and rim of the spheroids. In our case, we could not clearly identify this separation, particularly in control spheroids, where the transition between the two regions is largely faded. We can conclude that, in the case of GBM spheroids exposed to 1g, the central core region remains stable over time. In contrast, cells in the rim diffuse much less than those in the control experiment. This may indicate that the drug is unable to penetrate completely into the core of the spheroid; therefore, cells in the inner region continue to duplicate, while cells in the rim neither duplicate nor diffuse significantly. Moreover, the type of migration is different in the control case compared to the case with 1g, being mainly a collective and mesenchymal migration in the former case and an amoeboid and individual migration in the latter case.

Traction force microscopy experiments on U87MG spheroids revealed that an exposure to 1g resulted in a decreased contractility, which may contribute to a reduced invasion ability of cells within the 3D Matrigel matrix. Strong contractility is vital as it helps align fibers around the spheroid, allowing cells to exploit these fibers for directed migration into areas distant from the spheroid. The 1g exposure decreased the surface tension of cell aggregates (see Figure 3), and the individual behavior of cells in a 3D environment under 1g significantly diminished their contractility. This reduction hampered their interaction with extracellular fibers and their overall invasion potential. In contrast, the collective behavior of untreated U87MG cells led to a strong contraction of the extracellular matrix, which is positively correlated with their invasion ability [47]. Additionally, we assessed the mechanical properties of single cells using atomic force microscopy (AFM), observing an increase in cell rigidity under 1g conditions. In 2D culture analyses, 1g exposure considerably affected cell morphology, leading to a rapid loss of polarity (see Figure 4). In this case, it is important to consider that an increased contraction (or prestress) in cells in 2D typically leads to an increased rigidity of the cells as probed by AFM.

We also noted a clear correlation between the 1g concentration and the increased duration of cell mitosis (see Figure 5). Interestingly, the loss of polarity appeared to be cell-type dependent; for instance, the morphology of NIH-3T3 cells at the same 1g concentrations was unaffected. This loss of polarity correlated with a significant reduction in the migration area explored by U87MG cells in 2D cultures. We further investigated cell migration on substrates with varying rigidities. The mean square displacement (MSD) parameter indicated a biphasic behavior for U87MG cells in response to substrate rigidity, consistent with observations in other cell lines [69,92,93,94]. This behavior likely reflects a balance between cell adhesion to the substrate and the dynamics of actin polymerization and traction force generation [95,96]. On softer substrates, cells exhibited lower adhesion areas, producing a weaker attachment of focal adhesion complexes at the leading edge and an inefficient forward movement. Conversely, on very stiff substrates, the strong adhesion complexes development complicated the release of the cell’s rear portion during migration. Therefore, an intermediate substrate stiffness might provide an optimal balance, enhancing cell migration. A quantitative model based on the motor-clutch mechanism [69], originally designed to explain the traction force exerted by cells, can also be applied to the migration process. This model involves various modules oriented differently, each one connected to the central nuclear region, and analyzed according to the molecular clutch theory. These modules generate traction forces that facilitate effective cell migration [97]. The binding and unbinding rate constants, alongside the rate of the force application on the substrate, are fine-tuned to achieve an optimal intermediate substrate rigidity, ensuring that the rate of the force application neither hinders the formation of sufficient integrin/substrate attachment points before disengagement nor allows for a premature detachment of adhesion complexes before an adequate force is exerted on the substrate.

Whereas the motor-clutch mechanism primarily focuses on the roles of cytoskeletal actin polymers, adhesion complexes, and motor proteins, microtubules generally do not play a direct role. However, it has been shown that drugs affecting microtubule dynamics can also indirectly influence actin stress fibers, traction force, and cell migration [70,80,98,99]. Specifically, drugs that disrupt the microtubule attachment to integrin adhesion sites can activate Rho GTPase pathways, thereby strengthening actin stress fibers [100]. Furthermore, it has been demonstrated that drugs interacting with microtubules can alter the substrate rigidity value corresponding to the maximum explored area in cell migration, but, in our case, we did not see any appreciable variation of this value, probably because of a reduced resolution of the values of substrate stiffness that we considered. In some cases, research has shown that drugs interacting with microtubules and slowing their dynamics can reduce the sensitivity of cell migration to the stiffness of the substrate [101]. Our data suggest a similar trend: higher concentrations of 1g result in minimal differences in the area explored across various substrate rigidities. According to the molecular clutch model used for simulating cell migration, several critical parameters can influence cell migration behavior and its dependence on substrate rigidity. These include the number of molecular motors, the number of molecular clutches (particularly the ratio between the two), the actin polymerization rate (which guides pseudopodia formation and the capping of actin filaments), and the rate of pseudopodia formation. Previous studies have demonstrated that when drugs are used at concentrations that slow microtubule dynamics, there is a decrease in the actin polymerization rate and an increase in the pseudopodia nucleation rate. Both of these changes can lead to a reduction in the area explored by cells and a decreased sensitivity to the substrate’s Young’s modulus [101]. In our observations, particularly from the time-lapse videos of U87MG cells exposed to 1g (see Appendix A), we believe that the increased nucleation rate of pseudopodia is the most likely explanation for our findings. Each time the cell attempts to extend a pseudopodium, it fails to stabilize this extension, causing the cell to retract and initiate a new extension in a different direction.

Previous investigations into the effects of 1g have shown that it can influence microtubule dynamics, likely through the action of certain microtubule-associated proteins (MAPs) [40]. This mechanism suggests that 1g may have specific effects on different cell types, depending on the presence of various MAPs. In HeLa cells, the impact of 1g on the cytoskeletal structure during interphase was minimal, with its primary effects observed during the mitotic phase. Conversely, in U87MG cells, the loss of cell polarity induced by 1g indicates that it may also affect the microtubule dynamics during interphase, which contrasts with the behavior observed in HeLa cells. We can hypothesize that 1g, by acting on microtubule dynamics, inhibits the stabilization of processes after they are initiated. As a result, migrating cells may undergo rapid changes in their direction of motion, appearing similar to Brownian particles, even over short time intervals. It is also noteworthy to compare the observations made with 1g to the effects of taxanes on endothelial cells. In that context, a decrease in cell migration was noted, affecting both speed and directionality, at concentrations lower than those required to impair cell duplication [100]. The effect of taxanes was associated with an alteration in microtubule dynamics (the frequency of the transition from the growth to the shortening phase), resulting, at higher concentrations, in a strong decrease in the shortening rate and length, and a decrease in the number of cell adhesion complexes, accompanied by an enlargement of the remaining ones.

Taxol has been shown to reduce the presence of EB1-decorated plus ends of microtubules, which limits the delivery of the molecular components necessary for the protrusive activity of podosomes and their stabilization. This, in turn, affects cell polarization [101]. The stabilization of focal adhesion complexes is typically associated with the activity of the actin cytoskeleton, as actin filaments strongly interact with integrins, which are central to adhesion complexes. This stabilization is correlated with the formation of contractile bundles of actin and myosin motors, known as stress fibers. However, microtubules also target focal adhesion sites and interact with the cell cortex [102]. Microtubules dock at the plasma membrane through cortical microtubule stabilizing complexes (CMSCs) [103]. This association not only stabilizes microtubules against depolymerization but also promotes cell polarization. Consequently, there is a strong correlation between microtubule dynamics and the actin cytoskeleton, which can influence the formation or disassembly of adhesion complexes and cell protrusions. It is essential to consider that microtubules serve as pathways for the transport of molecules within the cell. Inhibitory signals are transported via microtubules, and if their dynamics is reduced, these signals may persist longer than necessary in the region where cell polarization begins, preventing its stabilization. This process could be the reason for the increase in the rate of the nucleation of processes but without their stabilization, as we reported on the basis of time-lapse imaging.

While many microtubule-targeting drugs are considered for their antimitotic effects, it has been reported that inducing complete antimitotic effects, which pushes cells toward apoptosis by attempting to override the spindle mitotic checkpoint, is not always necessary for these drugs to be effective. For example, paclitaxel has been shown to promote the formation of multipolar spindles at clinically relevant concentrations, leading to a majority of cells experiencing an abnormal exit from mitosis, a process known as mitotic slippage. This process results in gradual cell death during interphase [63]. One reason for this behavior may be that cells have access to more apoptotic pathways during the interphase compared to the mitotic phase, prompting them to undergo mitotic slippage to better engage these pathways. It has been hypothesized that paclitaxel targets the mechanisms responsible for correcting the frequent occurrence of multipolar spindles in cancer cells, particularly in cases of centrosome amplification [104]. Even in the case of 1g, we found that cells blocked in mitosis preferentially undergo a slippage to interphase, and they are typically killed after one attempt to perform mitosis. In general, given the increase of microtubule dynamics during mitosis, by altering microtubule dynamics, drugs can induce cells to form multipolar spindles even in the absence of centrosome amplification, consequently leading to a division with multipolar spindles [40]. However, microtubule-targeting drugs are also able to affect cell behavior during the interphase stage [64,105]. This impairment may occur because microtubules function as highways for transporting various signaling molecules, such as proteins, vesicles, and mitochondria. They play a crucial role in both inhibiting and promoting specific signaling pathways and interact with surface proteins on the cell. Several proteins play significant roles in the complex mechanism of cell migration, particularly within focal adhesion complexes. There are two competing effects related to the strength of focal adhesions in cellular motion: on one hand, the formation of larger structures is associated with a stronger adhesion to the substrate, leading to a reduced invasiveness. On the other hand, weak complexes are unable to generate the traction force necessary for cell migration [106]. In our study, we examined the expression of certain proteins present in focal adhesions through immunoblotting. Our results indicated that the expression of the structural protein α-actinin did not exhibit significant changes in the presence of 1g, with only a minimal decrease observed after 24 h of treatment. In contrast, the concentration of vinculin, a protein involved in mechanical force transduction, showed a significant reduction shortly after the initiation of 1g treatment. This finding agrees with earlier data indicating a loss of polarization within a short timeframe. Conversely, paxillin did not show any change in protein expression over the 24-h period compared to the control. Additionally, an analysis of focal adhesion kinase (FAK) protein expression revealed no differences at the tested time points. This suggests the possibility of an alternative pathway involved in pro-adhesion activity illustrated in Figure 6 and mediated by 1g. Notably, FAK does not interact directly with integrins but rather through adaptor molecules such as paxillin or α-actinin [107]. Previous research established that 1g can arrest the cell cycle in the G2 phase, significantly increasing p21 protein expression [60]. p21 is known to activate LIM kinase (LIMK), a family of actin-binding proteins that stabilize actin filaments by activating cofilin. Our findings showed that while Rac-1 expression is inhibited by 1g, LIMK levels were increased, which in turn deactivates cofilin through phosphorylation, an increase also observed after an exposure to 1g. Thus, we can speculate that the LIMK/cofilin activation may be triggered by p21. Moreover, since cofilin interacts directly with integrins through the c-Src complex [108], and is crucial for cell adhesion, the rise in the phosphorylated form of cofilin could partially account for the increase in cell adhesion and the decrease in cell motility induced by 1g. The evidence that 1g enhances cell adhesion while decreasing motility is critical regarding cancer cell spreading. Indeed, in various epithelial cancer cell lines, it has been demonstrated that poorly adherent cell populations tend to exhibit higher metastatic potential [109], suggesting that the adhesiveness of a particular tumor may serve as a physical marker for metastatic activity.

Interestingly, treatment with 1g increased the p-ERM/ERM ratio at both short and long incubation times. This behavior may indicate a feedback mechanism where the cortical actin cytoskeleton attempts to leverage the hyper-phosphorylation of ERM proteins to enhance contact with the membrane, resulting in an increased cortical tension. Simultaneously, ERM proteins crosslink microtubules to the plasma membrane. Research has shown that the destabilization of microtubules can trigger an almost instantaneous activation of ERM proteins, facilitating the attachment of cortical actin to the membrane [110]. This mechanism is responsible for the rounding of cells as they enter the mitotic phase and may explain the rounding of U87MG cells observed immediately after the exposure to 1g. In their study, Leguay et al. attributed the strong activation of ERM proteins to pathways associated with RhoA and the release of Ser/Thr kinases from the Ste20-like kinase family (SLK) [110]. The increase in the pERM/ERM ratio leads to an accumulation of p-ERM proteins near the plasma membrane and around the chromosomes during the mitotic phase. These findings agree with the results reported by Leguay et al. [110], where an increase in p-ERM, caused by the GEF-H1 release due to alterations in microtubule dynamics induced by nocodazole, resulted in a rounded cell shape, similar to what occurs during mitotic entry [111]. However, we established that the pathway promoted by compound 1g should differ from the effects of nocodazole. At the same time, it has been noted that taxol, a drug that stabilizes microtubules, does not induce an increase in p-ERM. The activation of ERM proteins is not solely dependent on their phosphorylation; rather, their active form is first achieved through an interaction with phosphatidylinositol 4,5-bisphosphate (PIP2), while phosphorylation later serves to stabilize this active form [112]. Once activated, ERM proteins connect actin to the plasma membrane by interacting with membrane receptors such as CD44. CD44 receptors are known to bind hyaluronan in the extracellular matrix and play a key role in cell migration and adhesion. Moreover, CD44 is typically overexpressed in cancer cells, which underscores the relevance of ERM proteins in the cell adhesion to substrates. The activation of ERM proteins by microtubule-destabilizing drugs has been reported to induce cell rounding during interphase, akin to the changes that occur when cells enter mitosis [113]. We can hypothesize a similar mechanism in the case of 1g, involving various proteins such as Rac-1, LIMK, cofilin, and ERM. In fact, Rac-1 has been shown to be crucial for microtubule polarity and, consequently, for proper cell motility and the functioning of kinetochores on mitotic chromosomes [114]. Additionally, Rac-1 is involved in cell cycle progression, promoting the transition from G2 into the mitotic stages [115]. We propose that the pathway of 1g action is closely related to the cortical layer of cells, where ERM proteins function, focusing on the connection between the cell membrane and cortical actin. This mechanism is mediated by microtubule dynamics and their association with cell adhesion complexes. For instance, it has been found that the activated and phosphorylated form of ERM proteins can interact with microtubules, stabilizing their connection with the cell membrane. This interaction may lead to the destabilization of focal adhesion complexes and hinder the cells’ ability to polarize [111]. A comparable situation was observed in migrating keratinocytes, where microtubule stabilization facilitated the disassembly of focal adhesion sites [116]. Moreover, the instability of microtubule connections with the cortical region of the cells is critical during the mitotic phase. Astral microtubules must maintain stable interactions with the cortical region to ensure a correct spindle positioning and the separation of centrosomes, or, in the cases of centrosome amplification, to cluster them effectively to form two opposing poles [117]. This mechanism may explain why 1g can influence the mitotic spindle in various cell types. The resulting effect of ERM activation is an increase in tension within the cortical layer. This observation is consistent with our findings from Micropipette Aspiration experiments. We noted that individual cells within the aggregates became more rounded and independent from one another after their exposure to 1g. This effect may explain the decreased surface tension observed in the treated spheroids. The interesting aspect of our analysis is that the effect of the compound 1g varies depending on the cell line being investigated, particularly concerning cell morphology and migration. In the case of NIH3T3 cells, we observed no significant changes in morphology induced by 1g, even at concentrations that led to a loss of polarity in U87MG cells. Additionally, the migration of fibroblasts was minimally affected by 1g. We speculate that this selectivity may be related to the role of microtubules and microtubule-associated proteins in different cell lines.

## 4. Materials and Methods

### 4.1. Cell Line and Treatment

The U87MG and NIH3T3 cell lines (ATCC), were cultured on polystyrene culture dishes (Euroclone, Pero, MI, Italy) and grown in EMEM, containing 1% non-essential aminoacids (NEA), 1% Na pyruvate, 1% glutamine, 1% pencillin/streptomycin, and supplemented with 10% FBS (all purchased from Sigma-Aldrich, Milano, Italy). The cells were kept in a humidified incubator at 37 °C with 5% CO_2_ until the time of experiments. A 1g compound was prepared in DMSO at 100 mM (stock solution) and diluted in appropriate media at the working concentrations [42,118,119]. The G166 isolated from malignant gliomas that show stem cell properties were a kind gift of Prof. Paolo Salomoni UCL Cancer Institute. The cells were grown in NS cell media (Stemcell Techologies, Vancouver, BC, Canada) added with human EGF and human FGF-2, 10 ng/mL of each, (Peprotech, Rocky Hill, NJ, USA) in a laminin-coated flask (Merck, Milano, Italy), and kept in a humidified incubator at 37 °C with 5% CO_2_.

### 4.2. 2D Cell Migration

A wound-healing assay was used to assess the effect of 1g on the cell migration of glioma cells. U87MG cell lines were seeded in multi-six-well plates and cultured until reaching confluence. A 10-µL pipette tip was used to make a straight vertical scratch. The cells were then treated with 1g (20 µM) or medium as a control. Images of the wounds were captured under an inverted light microscope Olympus IX 70 (10× magnification) by time-lapse microscopy for 24 h and measured using ImageJ software ver1.54k. Time-lapse imaging studies were performed using a phase contrast microscope with a home-developed on-stage cell incubator with a controlled temperature of 37 °C, a humidity of 90–95%, and 5% CO_2_ [120]. To perform single cell migration analysis, U87MG were plated at 3.5 × 10^3^ cells/cm^2^ in a plastic multi-six-well plate and in multi-six-well plates covered by a thin layer of PDMS to obtain substrates with a different value of the Young modulus: 0.2 kPa, 8 kPa, 32 kPa (CytoSoft^®^ 6-well Plates, Advanced Biomatrix, Carlsbad, CA, USA), and PDMS 1:25. The cells were incubated overnight and subsequently observed under the microscope (Olympus IX 70), at 10× magnification. The system takes a photo every 4 min. This system allowed us to simultaneously evaluate the effects of 1g at three different concentrations: 5 μM, 10 μM, and 20 μM. The time-lapse videos were then edited using ImageJ and cell positions were recorded using the manual tracking software of FIJI ver. 2.16.0.

### 4.3. Adhesion Assay

U87MG cells pre-treated (6 h with 1g at 20 μM) or not were grown in 96-well plates pre-coated with BME 0.25% at 90,000 cells/cm^2^, and incubated or not with 1g at 5, 20, and 50 μM, in serum-free EMEM, and incubated at 37 °C for 30 min, followed by a wash with phosphate-buffered saline (PBS) to remove non-adherent cells. The cells were fixed in cold acetone for 15 min and stained with 0.5% crystal violet for 1 h and read at 540 nm using a spectrophotometer (Thermofisher, Biorad, Bologna, Italy).

### 4.4. Western Blot

Proteins were extracted from the U87MG control and 1g-treated cells (20 μM) using RIPA buffer (50 mMTris-HCl pH 7.4, 150 mMNaCl, 1% Na deoxycolate, 1% Triton X-100, and 2 mM PMSF) (Sigma Aldrich, Milano, Italy). Lysate proteins were quantified using the BCA Protein Assay Kit (Life Technology, Milano, Italy) according to the manufacturer’s protocol. A total of 5 mg of each sample was loaded onto a pre-cast 5–12% SDS-PAGE (Invitrogen, Milan, Italy) and transferred to a nitrocellulose membrane (Invitrogen, Milan, Italy). The membrane was blocked in a TBST (20 mMTris- HCl, 0.5 M NaCl, and 0.05% Tween 20) buffer containing 5% non-fat dried milk overnight at 4 °C and incubated with the primary antibody anti-ɑ-actinin (1:1000), anti-paxilin (1:1000), anti-phospho-Erm (phospho-ezrin (Thr567)/radixin (Thr564)/moesin (Thr558) (1:1000), anti-Erm (1:1000), and anti-vinculin (1:1000), anti-phospho-Fak (Y397) (1:1000), anti-Fak (1:1000), anti-RhoA (1:1000), and anti-Rac1 (1:1000) at RT, respectively, for 3 h under gentle agitation (the primary antibodies were from Cell Signaling, Danvers, MA, USA). After being washed in TBST, the membranes were incubated for 1h with an HRP-conjugated anti-rabbit antibody (Cell Signaling, USA) or with a HRP-conjugated anti-mouse antibody (Cell Signaling, USA) and visualized using the chemiluminescence method (Amersham, GE Healthcare Europe GmbH, Milan, Italy). The immune complexes were analyzed using the Imagestudio lite software 4.0 C-Digit with β-actin as the loading control.

### 4.5. Spheroid Preparation

For the formation of spheroids, the U87MG cells were seeded in the 96-well plates ultra-low attachment (Costar, Corning Inc., Corning, NY, USA) at a density of 500 cells/well in a complete medium.

The following day, the spheroids were treated with 1g 20 µM and after 24 h analyzed with 3D traction force microscopy and Micropipette Aspiration. For the analysis of spheroid invasion, spheroids were embedded in a Corning^®^ Matrigel^®^ gel (10 mg/mL) (Corning Inc., New York, NY, USA) and their evolution was measured by optical microscopy in bright contrast acquiring an image every hour. The best focused plane was acquired using a 10× objective and the resulting image is obtained stitching 4 fields together in order to capture the whole region of interest For 3D traction Force Microscopy, to assemble the system, we mixed the beads with the Matrigel^®^ and we deposited a layer of the solution in a well of a 96-well-plate. We then waited for the Matrigel^®^ to start the polymerization and we subsequently deposited a spheroid on it. The spheroid was then covered with another layer of the same Matrigel^®^, a layer of the medium, and we waited another time interval for the polymerization of the newly deposited gel before starting the acquisition of the images.

### 4.6. The 3D Traction Force Microscopy of Spheroids

The measurements of the traction forces exerted by the spheroids upon the surrounding micro-environment has been evaluated following the protocol developed by ref. [48] and previously by ref. [51].

First, the time-lapse images of the spheroids have been aligned to a given frame of reference. Then, the field of displacements (deformations) around the spheroids has been evaluated adopting the PIV method. We used a square window size of 40 pixels with a cut-off of 650 pixels (1 pixel = 0.73 μm since the images had been captured using a 20× magnification). The PIV method compares a couple of subsequent images of the sequence identifying the spheroid boundaries, the field displacements (deformations), and eventually the drift correction can be included. The field of displacements are stored in a dedicated subdirectory in .npy python arrays.$$\kappa \left(\varepsilon \right)=\kappa_{0}\cdot \left\{\begin{array}{c} e^{\frac{\varepsilon }{d_{0}}}\;for\;\varepsilon <0\;bucling\\ 1\;for\;0<\varepsilon <\varepsilon_{s}\;linear\;regime\\ e^{\frac{\left(\varepsilon -\varepsilon_{s}\right)}{d_{s}}}\;for\;\varepsilon \;\geq \varepsilon_{s}\;strain\;stiffening \end{array}\right.$$

Before the final force reconstruction step, the lookup table must be evaluated. The lookup table describes the connection between the normalized distance (*r*/*r*_0_) and the normalized deformation (*d*/*r*_0_) for different applied pressures. The previous authors [48] provide a list of several lookup tables for Matrigel^®^ and collagen with different densities. We evaluated the appropriate lookup table for Matrigel^®^ 10 mg/mL using the semi-affine material model previously exploited by ref. [51]. This model is able to simulate a non-linear material subdividing behavior in three different regimes: buckling, linear regime, and strain stiffening.

The ε parameter represents the strain, and κ_0_ is the linear stiffness, while d_0_ and d_s_ are the rate of stiffness change during the buckling and stiffening phase, respectively. Finally, the ε_s_ parameter is the onset of strain stiffening. During the buckling regime, the material is under compressions (negative strain) and the fibers constituting the material are not able to respond to the external stimuli, and the stiffness decreases exponentially. For limited positive strains, the material produces a linear response. For larger positive strains (above *ε_s_*), the material undergoes strain stiffening. Since *d*_0_, *d_s_*, and *ε_s_* are independent with respect to the particular Matrigel^®^ concentration, we need to adjust only the linear stiffness parameter before proceeding with the lookup table evaluation.

After obtaining the lookup table for Matrigel^®^ with a 10 mg/mL concentration, we can pass to the force reconstruction step. To avoid artifacts due to cells being close to spheroid boundaries, we reconstruct the force for distances larger than 2 spheroid radii. The results are summarized in an .xls file containing the fitted pressure, the measured contractility, and their angular distributions.

### 4.7. Micropipette Aspiration Technique

The microaspiration of cell spheroids were performed exploiting a micropipette prepared by pulling borosilicate capillaries (World Precision Instruments, WPI, Sarasota, FL, USA), 1.5 mm/1 mm O/I diameter) with a double step pulling strategy. The micropipettes were initially thinned using an automatic puller and then were further thinned using a custom-developed puller. A custom-developed forge was exploited to cut pipettes in order to have the pipette aperture perpendicular to its longitudinal. Pipettes were fire-polished to ensure good contact with the spheroid and to avoid damage to cells in the periphery of the spheroid. To prevent adhesion between the glass sides of the pipette and cells, the micropipettes were pretreated with BSA (10 mg/mL) or Surfasil. In the case of the pretreatment with BSA, the pipettes were immersed in the BSA solution for 5 min and then they were thoroughly rinsed with distilled water before being filled with the same culture medium used to grow the spheroids. In the case of the Surfasil pretreatment, the micropipettes were immersed for 5 min in a toluene-diluted Surfasil solution and they were subsequently thoroughly washed with toluene and then kept for 10 min in the oven at 90 °C to allow for a complete adhesion of the layer in contact with the pipette glass. Each pipette was then connected to a pneumatic pressure transducer (Lorenz MPCU-3, sensitivity of 1 mm H_2_O) to establish a pressure difference between the internal side of the pipette and the external solution. The pressure difference was applied by controlling the air pressure on top of a cylindrical tube containing the culture medium solution. The tube was initially positioned in order to assure the position of the free surface of the culture medium that allowed a negligible starting pressure difference, verified by controlling the null aspiration or repulsion of small objects in the solution. The spheroids were kept inside a chamber made by glass slides separated by a PDMS or Teflon ring allowing for the entry of the pipette from one side and the injection of a single spheroid from the other side. The bottom glass of the chamber was pretreated with BSA or Surfasil to avoid adhesion with the spheroid during the time needed to find and grab it with the micropipette. We performed a creep compliance analysis in the time domain analyzing the response of the spheroid inside the micropipette after a pressure difference jump of about 20 cm H_2_O (between the internal pipette region and the region just outside the pipette). To analyze the creep behavior, the progressive position of the spheroid protrusion inside the micropipette is measured for 20 min in the aspiration and another 20 min in the release phase by exploiting optical microscopy images and at a rate of 1 frame per min. Images were acquired by an Olympus IX 70 inverted microscope in Differential Interference Contrast (DIC) mode with a 20× or 40× objective. To maintain, as much as possible, optimal conditions for cells, a closed system was used to inject humidified water vapor at a temperature of 37 °C with a CO_2_ concentration of 5% (see Supplementary Materials of ref. [121]). To confirm the state of cells, a LIVE/DEAD cell imaging Kit has been used to establish cell viability on the basis of intracellular esterase activity and plasma membrane integrity (two different filters, FITC and TRITC were used as follows: green → live cells, red → dead cells) (Appendix A). The images were then analyzed by using the ImageJ software in order to automatically detect the position of the cell protrusion inside the micropipette.

According to the model reported in Figure 3c, the specific behavior for the creep response of the spheroid can be written as follows [56]:(2)$$L\left(t\right)=\frac{F}{k_{1}}\left(1-\frac{k_{2}}{k_{1}+k_{2}}e^{-\frac{k_{1}k_{2}}{\eta_{2}\left(k_{1}+k_{2}\right)}t}\right)+\frac{F}{\eta_{1}}t$$
where *L*(*t*) is the spheroid projection inside the micropipette and *F* is the applied stress, which in this case is given by the following:$$F=∆P-∆P_{c}$$
where Δ*P* is the effective pressure step that is applied and it represents the parameter we can control, and Δ*P_c_* is the critical pressure, i.e., the pressure above which the spheroid starts to move inside the micropipette. In Equation (1), we can identify a time constant, i.e., a characteristic viscoelastic time for the initial spheroid deformation, given by the following:$$\tau_{c}=\frac{\eta_{2}\left(k_{1}+k_{2}\right)}{k_{1}k_{2}}$$

Moreover, the equation for *L*(*t*) can be considered as the sum of a first contribution due to an initial viscoelastic behavior of the spheroid, and a second contribution that corresponds to a continuous flow of the spheroid inside the micropipette at a constant velocity given by the following:$$\frac{dL(t)}{dt}\left(t\to \infty \right)=\frac{F}{\eta_{1}}$$

The dissipative term connected to Equation (1) is related to the rearrangement of the cells composing the spheroid when they cross the initial mouth of the micropipette. In fact, a good passivation of the micropipette inner surface with BSA or Surfasil should avoid the presence of viscous dissipation due to the spheroid flowing inside the micropipette in contact with its inner wall. According to Equation (1), we have the initial tongue deformation:$$L\left(0\right)=\frac{F}{k_{1}+k_{2}}$$

The meaning of the two spring constants, *k*_1_ and *k*_2_, can be associated with two Young moduli, which are related to the elasticity of the spheroid which comes into play at long times (with respect to a characteristic time) after the start of the aspiration process (*k*_1_) and to the instantaneous elastic behavior (*k*_2_) of the actin cortex of the cells.

Δ*P_c_* can be related to the Laplace law for the aspirated spheroid with a surface tension γ. In fact, we can write the following:$$∆P_{c}=2\gamma \left(\frac{1}{R_{p}}-\frac{1}{R}\right)$$
where *R_p_* is the radius of the micropipette and *R* is the radius of the spheroid. Experimentally, Δ*P_c_* could be measured by considering the minimum applied pressure difference that initiates the continuous flow of the spheroid tongue inside the pipette. In practice, this is the Laplace pressure due to the curvature imposed by the pipette size.

The Δ*P_c_* term can be evaluated following another procedure. By neglecting the initial part of the spheroid deformation, in the aspiration part of the experiment we can measure the constant flow speed of the tongue inside the micropipette at long times. This value should be given by the following:$$v_{in}=\frac{dL(t)}{dt}=\frac{F}{\eta_{1}}=\frac{∆P-∆P_{c}}{\eta_{1}}$$
If the external pressure is released, the pressure driving the tongue retraction is given by $F=-∆P_{c}\;$and we have, after a long enough time interval:$$v_{out}=\frac{dL(t)}{dt}=\frac{F}{\eta_{1}}=\frac{-∆P_{c}}{\eta_{1}}$$
If we consider the two speeds we just calculated, we can obtain the following:$$∆P_{c}=∆P\frac{-v_{out}}{v_{in}-v_{out}}$$
where we have to consider that the retracting speed is negative.

For sufficiently long times since the initial pressure difference step, Equation (2) becomes the following:$$L\left(t,t\to \infty \right)=\frac{F}{k_{1}}+\frac{F}{\eta_{1}}t$$

Accordingly, a linear fit to the last part of the aspiration data provides both the speed of the entering spheroid (*v_in_*), as we previously described, and the intercept for *t* = 0 that corresponds to $\delta =\frac{F}{k_{1}}$. At the same time, considering the stress/strain relationship, the first term of Equation (2) represents the typical behavior of a viscoelastic solid body with the first Young modulus given by the following:$$\frac{F}{\pi R_{p}^{2}}=E\frac{L}{R_{p}}\;\to \;L=\frac{F}{\pi R_{p}E}\;k_{1}=\pi R_{p}E\;\to \;E=\frac{k_{1}}{\pi R_{p}}$$
which comes into play for times longer than the characteristic one defined by the exponential dependence. Accordingly, for the Young modulus, we obtain the following:$$E_{1}=\frac{F}{\pi R_{p}\delta_{1,control}}=\frac{R_{p}\left(∆P-∆P_{c}\right)}{\delta_{1,control}}$$
Considering Ref. [59] and the fact that we functionalized the inner surface of the glass pipette to avoid adhesion as much as possible, we can calculate the viscosity of the aggregates according to the following:$$\eta =\frac{R_{p}\left(∆P-∆P_{c}\right)}{3\pi \frac{dL\left(t\right)}{dt}(t\to \infty )}$$

### 4.8. Dynamic Mechanical Analysis by AFM

Cells were analyzed with a BioScope I microscope equipped with a Nanoscope IIIA controller (Veeco Metrology, Plainview, NY, USA). The cantilever spring constant has been calibrated using the thermal noise method [122]. Dynamic mechanical measurements have been performed exploiting a home-developed device that can apply a sinusoidal signal taken from a lock-in amplifier to the z-piezo-scanner of the AFM obtaining a modulation of the cantilever position when a constant indentation is maintained. The other input of the lock-in amplifier is connected to the signal of the cantilever deflection detector. The lock-in amplifier detects the amplitude of the deflection signal and the phase lag between the two signals. Due to the low integration time of the lock-in amplifier and to stability problems of the AFM cantilever, we considered frequencies not lower than 1 Hz (enough cycles must be obtained before the signal stabilizes). When the cantilever oscillates around an indentation δ_0_, we can write the following equation [79]:$$F\left(\omega \right)=2\frac{E^{*}\left(\omega \right)}{1-\nu^{2}}\sqrt{R\delta_{0}}\delta (\omega )$$
where R is the radius of the indenting probe, and *F*(*ω*) and *δ*(*ω*) are the periodic force applied by the tip and the sample indentation, respectively. The indentation is obtained by subtracting the cantilever deflection from the piezo vertical displacement. The term *E** is the complex Young modulus in the frequency domain and ν is the Poisson ratio. The previous equation is related to the Hertz theory of contact mechanics, whereas rheological investigations typically consider the shear modulus *G*. It is possible to consider the parameter G according to the transformation *G* = *E*/2(*ν* + 1). The correspondence principle for linear viscoelasticity allows us to write the following:$$G^{*}\left(\omega \right)=\frac{1-\nu }{4\sqrt{R\delta_{0}}}\frac{F\left(\omega \right)}{\delta (\omega )}$$
where *F*(*ω*) and *δ*(*ω*) are the Fourier transforms of the force and indentation parameters and *G** is the complex shear modulus. The complex modulus can also be written as follows:$$G^{*}\left(\omega \right)=G^{\prime }\left(\omega \right)+iG^{\prime \prime }\left(\omega \right)$$
where $G^{\prime }\left(\omega \right)$ is the shear storage modulus (related to elastic behavior) and $G^{\prime \prime }\left(\omega \right)$ is the shear loss modulus (related to viscous dissipation). *F*(*ω*) and *δ*(*ω*) can be written as follows:$$F\left(\omega \right)=F_{0}\left(\omega \right)e^{i\vartheta (\omega )}\;\delta \left(\omega \right)=\delta_{0}\left(\omega \right)e^{i\phi (\omega )}$$
and their ratio is given by the following:$$\frac{F_{0}\left(\omega \right)}{\delta_{0}\left(\omega \right)}e^{i[\vartheta \left(\omega \right)-\phi \left(\omega \right)]}=\frac{F_{0}\left(\omega \right)}{\delta_{0}\left(\omega \right)}e^{i\Delta \theta (\omega )}$$

Considering the expression for the complex shear modulus, we can write the following:$$G^{\prime }\left(\omega \right)=\frac{1-\nu }{4\sqrt{R\delta_{0}}}\frac{F_{0}\left(\omega \right)}{\delta_{0}\left(\omega \right)}\cos\Delta \theta (\omega )\;G^{\prime \prime }\left(\omega \right)=\frac{1-\nu }{4\sqrt{R\delta_{0}}}\frac{F_{0}\left(\omega \right)}{\delta_{0}\left(\omega \right)}\sin\Delta \theta (\omega )\;$$

The phase lag Δθ(ω) between the applied force and the cantilever deflection, equivalent to the produced indentation, measured by the AFM detector could be affected by spurious phase lags due to the electronic system (mainly the piezoactuator device) and to the viscous drag of the cantilever when excited to high frequencies. To account for the first spurious contribution, we performed measurements of the phase lag between the two quantities when the cantilever was in contact with a rigid and not deformable surface like mica. Considering that the phase lag observed in this case is not due to a viscous behavior of the indented sample, this value of the phase lag was subtracted from the value we obtained on living cells at different frequencies. To account for the drag force, we included in the formula a dissipative term:$$G^{*}\left(\omega \right)=\frac{1-\nu }{4\sqrt{R\delta_{0}}}\left(\frac{F\left(\omega \right)}{\delta \left(\omega \right)}-i\omega b(0)\right)$$$$G^{\prime \prime }\left(\omega \right)=\frac{1-\nu }{4\sqrt{R\delta_{0}}}\frac{F_{0}\left(\omega \right)}{\delta_{0}\left(\omega \right)}\sin\Delta \theta (\omega )-\omega b(0)$$
and for *b*(0) we took the value reported in the literature for a similar cantilever (*b*(0) = 5 × 10^−6^ Ns/m) [78,123].

### 4.9. A Ting Model Analysis of Single Force Curves

An extensive rheological analysis of living cells can be realized also considering viscoelastic models to describe the complete approach and retract portions of the force curve. To perform this analysis, we exploited the Ting model using a procedure similar to the one presented in the work by Efremov et al. [83]. Briefly, the Ting solution for a rigid pyramidal indenter exerts a force on a viscoelastic material is exploited according to the below formulas:(3)$$\left\{\begin{array}{c} F\left(t,\delta \left(t\right)\right)=\frac{1.4906\tan\vartheta }{2(1-\nu^{2})}\int_{0}^{t}E\left(t-\xi \right)\frac{\partial \delta }{\partial \xi }d\xi \;for\;0\leq t\leq t_{m}\\ F\left(t,\delta \left(t\right)\right)=\frac{1.4906\tan\vartheta }{2(1-\nu^{2})}\int_{0}^{t_{1}(t)}E\left(t-\xi \right)\frac{\partial \delta }{\partial \xi }d\xi \;for\;t_{m}\leq t\leq t_{ind}\\ with:\;\int_{t_{1}(t)}^{t}E\left(t-\xi \right)\frac{\partial \delta }{\partial \xi }d\xi =0\;for\;t_{m}\leq t\leq t_{ind} \end{array}\right.$$
where *F* is the tip/sample force*, t_ind_* is the time of the entire indentation cycle (approach and retract phases), *t_m_* is the inversion time of the indentation cycle, and *t*_1_ is a parameter needed to compensate for the sample relaxation during the retraction phase and it is calculated by the expression reported in the above formula, and *E*(*t*) represents the Young’s modulus relaxation expression. For *E* we adopted the following expression:$$E\left(t\right)=E_{0}{\left(\frac{t}{t\prime }\right)}^{-\alpha }$$
where *t*′ corresponds to the sampling time during the force curve, and *E*_0_ is the instantaneous Young modulus of the sample. The adopted relaxation expression is known as the power law rheology model and it is the typically exploited model for AFM rheological measurements [83]. Equation (3) is numerically solved and a least squares error fitting procedure using the experimental values of the force is exploited to find the best values for the two parameters E_0_ and α. All the force curve analyses were performed with a home-developed Python software (ver. 3.7). The vertical scanning tip speed was 8 μm/s, and the total z-scan displacement was between 2.5 μm and 3 μm. The fitting procedure was repeated for all the sample points of the Force Volume image (32 × 32 pixels^2^) and the average values from force curves obtained over the nucleus region were evaluated. The cells were imaged for a total time of about 3 h while keeping the temperature constant at 37 °C.

## 5. Conclusions

In conclusion, we have established the ability of the molecule 1g to interfere with cell adhesion, motility, spreading, and the mitotic stage of glioblastoma multiforme cells. Although the specific targets of this action are not yet fully understood and require further analysis, we hypothesize that its activity is related to microtubule dynamics, likely through its interaction with certain microtubule-associated proteins (MAPs). The effect of this molecule on microtubules influences the stability of focal adhesion complexes, leading to a rapid rounding of cells, while leaving the structure of the microtubules largely intact. This is consistent with the role of microtubules in establishing cell polarization. Cell rounding is associated with the activation of ERM proteins, which increases cortical tension within the cells. Changes in the cortical layer may in turn affect the organization of the mitotic spindle, potentially resulting in the formation of multipolar spindles and, ultimately, leading to mitotic slippage before cell apoptosis.

Although this research is focused on in-vitro models and further studies are necessary to identify and understand the specific molecular pathways involved in the activity of this novel 2,3-benzodiazepine derivative, the compound 1g has already demonstrated promising pharmacological characteristics. These features make it a strong candidate for future research and development within the field of cancer research. Moreover, the potential applications of compound 1g may extend into cancer therapy itself, offering hope for new treatment options. As the scientific community continues to explore the intricate mechanisms of this compound, it is essential to investigate its efficacy and safety further, paving the way for advancements in cancer treatment strategies.

## Figures and Tables

**Figure 1 ijms-26-02767-f001:**
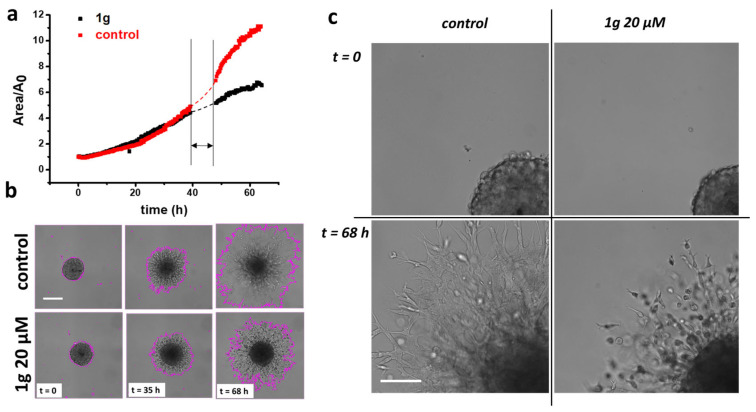
(**a**) A normalized equatorial area expansion of a control (DMSO) U87MG spheroid (black curve) and a spheroid of the same cells treated with 20 μM 1g (red curve). The spheroids were embedded inside a Matrigel^®^ matrix as described in Section 4: Materials and Methods; (**b**) images of a control and of a 1g-treated spheroid at different time points. The line highlighting the spheroid expansion has been obtained using the Analyze_Spheroid_Cell_Invasion_In_3D_Matrix tool from FIJI (bar = 200 μm for all six images). (**c**) A comparison of the cell shape between the control sample and the U87MG spheroid exposed to 20 μM 1g. The frames are representative of the spheroids at the beginning of the experiment and after 68 h. (bar = 50 μm for all four images).

**Figure 2 ijms-26-02767-f002:**
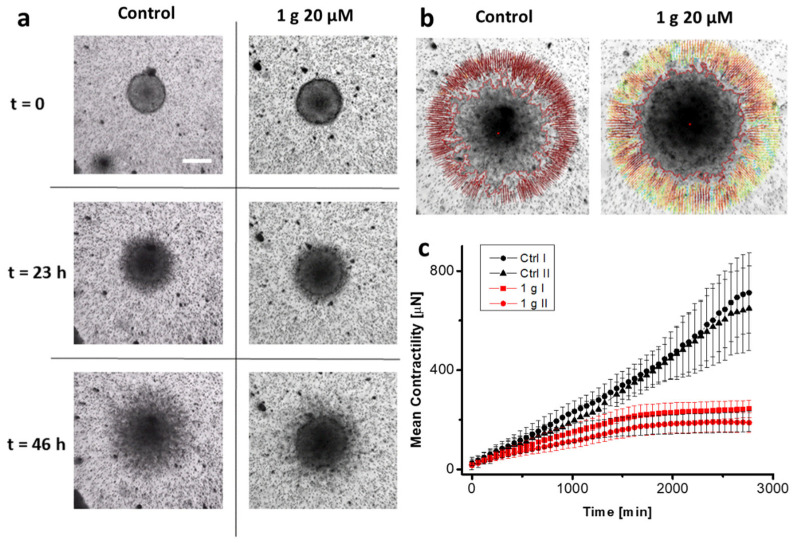
The traction force microscopy of U87MG spheroids. (**a**) Frames at different time steps of a control U87MG spheroid and of a U87MG spheroid in 15 μM 1g. The spheroids were embedded in a Matrigel^®^ matrix (10 mg/mL protein content, Young’s modulus of about 400 Pa) with 5 μm diameter latex beads (bar = 200 μm for all six images); (**b**) a representation of the traction force at the end of the experiment for a control and 1g-treated spheroids. The arrows show the direction and intensity of the bead displacement; (**c**) a plot of the mean contractility as a function of time for spheroids in the different conditions (two experiments for each of the conditions are reported).

**Figure 3 ijms-26-02767-f003:**
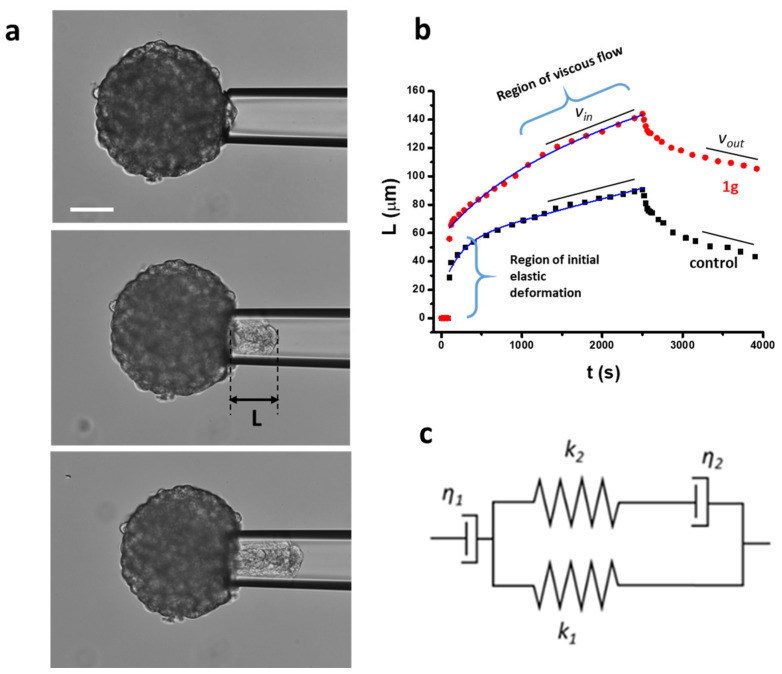
The micropipette aspiration of U87MG spheroids. (**a**) The sequence of three images during the aspiration process of an U87MG spheroid exposed to 20 μM 1g. *L* represents the length of the tongue to be used for the fitting procedure (bar = 50 μm for all images); (**b**) the representative aspiration and relaxation curves for a control spheroid (black squares) and a 1g-treated spheroid (red circles). In the plot, the regions exploited to obtain *v_in_* and *v_ou_*_t_ (see text) are highlighted. The continuous lines represent the fit of Equation (1) to the data; (**c**) the rheological model exploited to obtain Equation (1). According to this model, in parallel with a Voigt element (a spring characterized by a Young modulus k_2_ in series with a damper η_2_ representing a dissipation coefficient related to the initial instantaneous spheroid deformation) there is a spring with Young modulus k_1_ to account for the instantaneous elastic deformation (the region is highlighted in the plot) and the overall model is in series with another dashpot η_1_ (viscous dissipation of the spheroid tongue) to account for the long-term flow of the cell aggregate.

**Figure 4 ijms-26-02767-f004:**
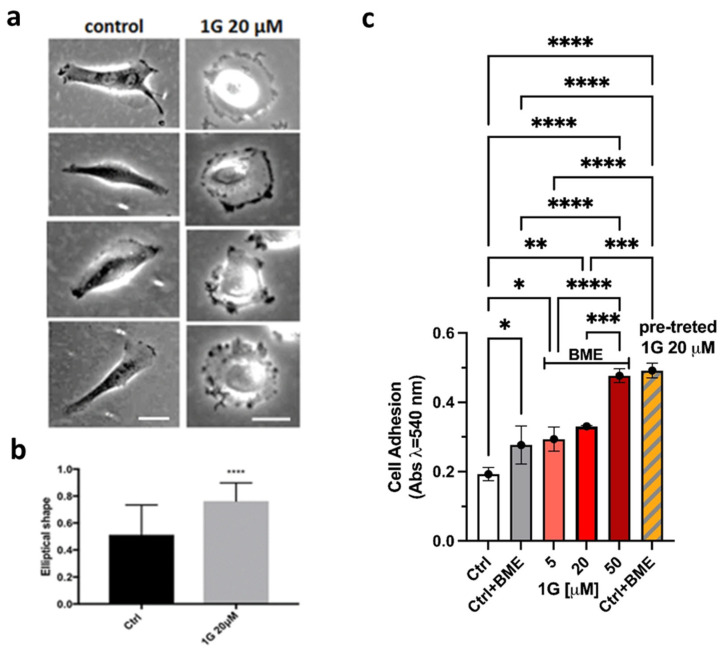
The effect of 1g on the morphology and adhesion of U87MG cells. (**a**) Examples of U87MG cells before the injection of 20 μM 1g into the culture medium and their corresponding morphology 1 h after the injection (bar = 20 μm in each column of images); (**b**) a statistical analysis of the elliptical shape defined as the ratio of the two axes resulting from an elliptical fit of the cell morphology; (**c**) a cell adhesion assay for the U87MG cell line. Cells, pre-treated or not with 1g 20 μM, were grown on BME and incubated with EMEM without FBS alone (Ctrl and pre-treated with 1g 20 μM) or with 5, 20, or 50 μM of 1g and expressed as rates of U87MG cell adhesion to culture plates (abs = 540 nm). **** *p* < 0.0001, *** *p* < 0.001, ** *p* < 0.01 and * *p* < 0.05 vs. respective; one way ANOVA and Tukey’s as post test.

**Figure 5 ijms-26-02767-f005:**
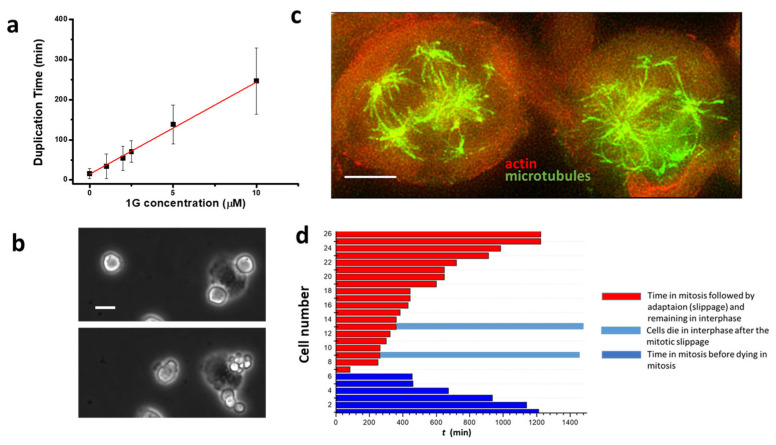
The effect of 1g on cell duplication. (**a**) A plot of the duplication time (measured as the time cells remain rounded in mitosis) as a function of the 1g concentration. The red line represents a linear fit to the data. (**b**) An example of cells exiting mitosis and producing more than two daughter cells (bar = 20 μm for both images); (**c**) examples of the presence of more than two asters in the organization of microtubules in mitosis in the presence of 20 μM 1g. The actin cytoskeleton has been marked in red and the microtubules in green (bar = 5 μm); (**d**) a time-lapse imaging analysis of U87MG cells exposed to 20 μM 1g in mitosis. For 26 cells observed by the time lapse-imaging, the time spent in mitosis has been recorded. The large majority (78%) of the cells exposed to 20 μM 1g are able to perform mitotic slippage.

**Figure 6 ijms-26-02767-f006:**
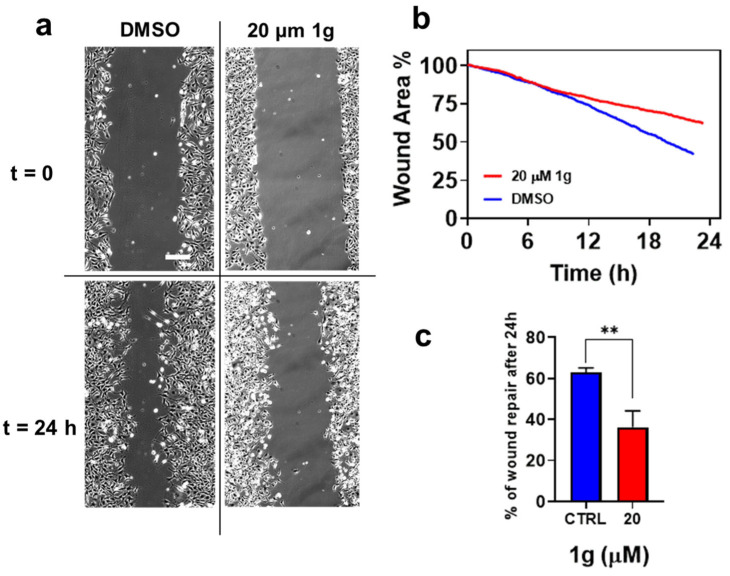
Wound healing assay for control cells and cells exposed to 1g. (**a**) An optical microscopy image of a 2D wound healing assay immediately after the formation of the scratch and after 24 h since the formation of the wound both in the absence and in the presence of 20 M 1g (bar = 200 μm in all four images); (**b**) the time evolution of the still-present wound area; and (**c**) the percentage of the wound repair closure after 24 h in the absence of 1g and in the presence of 20 μM 1g (*p* = 0.0049 (**), unpaired *t* test).

**Figure 7 ijms-26-02767-f007:**
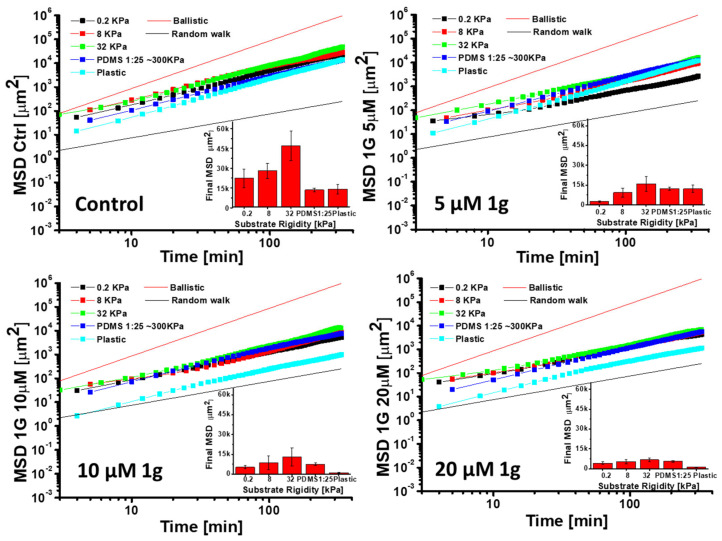
An analysis of the MSD of U87MG cells on substrates of different Young moduli and exposed to increasing concentrations of 1g. For the migration analysis, we considered cells that did not interact by physical contact with other cells and we stopped tracking the cells once they reached the typical globular configuration of the mitotic stage. The MSD vs. time is reported in a Log–Log plot with two lines representing the ballistic and the pure Brownian motion cases. In the inset of each plot, the MSD value at the end of the corresponding plot is reported as a function of the substrate stiffness. The plots are limited to the time intervals for which the standard deviation of the data does not become too large (data for the longest time intervals are averaged over a smaller set of measurements). The presence of a maximum value of the MSD for an intermediate value of the substrate stiffness for the control case is evident. By increasing the 1g concentration, the sensitivity of the MSD value to the substrate stiffness strongly decreases.

**Figure 8 ijms-26-02767-f008:**
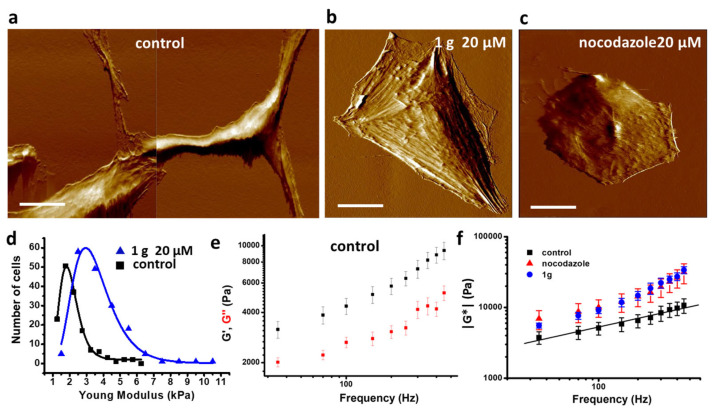
Single-cell mechanical properties measured by AFM: (**a**) an AFM image of control U87MG cells; (**b**) an AFM image of a U87MG cell exposed to 1g 20 μM for 24 h; (**c**) an AFM image of a U87MG cell exposed to nocodazole 20 μM for 24 h. (bar = 20 μm for all AFM images); (**d**) distributions of the values obtained for the Young modulus of U87MG cells before (black squares) and after 24 h in 20 μM 1g (blue triangles). The points in the plot are derived from the distribution histogram of the value. The Young modulus was obtained by fitting the force curves with a modified Hertz model; (**e**) elastic (G′, red squares) and viscous (G″, black squares) components of the shear modulus G* as a function of the frequency of the sinusoidal signal applied to the cantilever base by the piezoactuator in control U87MG cells; and (**f**) a comparison of the modulus of the complex shear modulus G*, as a function of frequency, of U87MG cells for the untreated, exposed to 20 μM 1g for 24 h and exposed to 20 μM nocodazole for 24 h. Data represent the average value and the standard deviation of *n* = 7 cells for each condition. The slope of the linear fit to the control cell data is 0.33.

**Figure 9 ijms-26-02767-f009:**
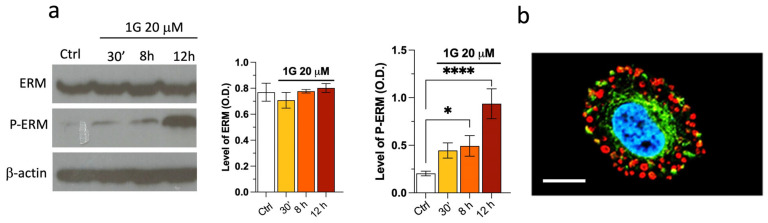
(**a**) The expression of ERM and pERM proteins in U87MG cells exposed to 20 μM 1g. The analysis shows the strong increase of the pERM/ERM ratio; (**b**) the immunofluorescence of typical U87MG cells 30 min after the injection of 20 μM 1g: pERM (red), microtubules (green), and DNA (blue) (bar = 20 μm). **** *p* < 0.0001 and * *p* < 0.05 vs. Control; one way ANOVA and Tukey’s as post test.

**Figure 10 ijms-26-02767-f010:**
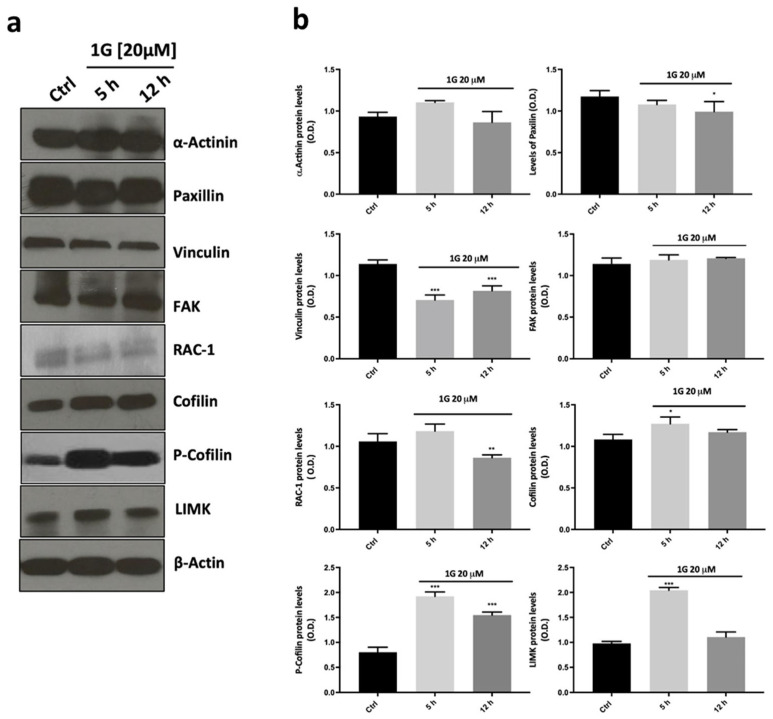
The effect of 1g on proteins involved in cell adhesion and motility. (**a**) Representative western blots of a-actinin, paxilin, vinculin, FAK, Rac-1, cofilin, P-cofilin, and LIMK at different time points; (**b**) a densitometric evaluation of protein levels in U87MG cell lysate after incubation with 20 μM of 1g for 5 or 12 h. Densitometry values were normalized to the protein loading control, beta-actin. The values are expressed as the mean ± SD of three independent experiments (*n* = 3 per group). *** *p* < 0.001, ** *p* < 0.01 and * *p* < 0.05 vs. untreated cells (Ctrl), using one-way ANOVA with Dunnett’s as a post test.

## Data Availability

The data presented in this study are available on request from the corresponding authors.

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
