# Peer review of "A Benzodiazepine-Derived Molecule That Interferes with the Bio-Mechanical Properties of Glioblastoma-Astrocytoma Cells Altering Their Proliferation and Migration"

_ijms, 2025, doi:10.3390/ijms26062767_

Round 1
Reviewer 1 Report
Comments and Suggestions for Authors
Strengths of the Manuscript:
-
Scientific Merit: The study presents a novel benzodiazepine-derived molecule (1g) that potentially affects glioblastoma cell proliferation and migration, contributing valuable insights into glioblastoma treatment.
-
Comprehensive Methodology: The use of multiple experimental techniques, including micropipette aspiration, traction force microscopy, and 3D spheroid invasion models, provides a robust investigation of the effects of 1g.
-
Mechanistic Insights: The manuscript effectively discusses the potential mechanisms of action, highlighting the role of microtubule-associated proteins (MAPs) and the impact of 1g on cellular biomechanics.
-
Clinical Relevance: The discussion regarding the blood-brain barrier penetration of 1g suggests potential translational applications.
Major Revisions:
-
Introduction Clarity:
-
The end of the introduction contains excessive details regarding the study's results. While it is important to summarize the significance of the research, these findings should be relocated to the Discussion or Results section to enhance the logical flow.
-
-
Selection of 15-20µM Concentration:
-
The manuscript lacks an IC50 curve for 1g, making it unclear why the authors selected 15-20µM. The inclusion of such data would strengthen the justification for the chosen concentration range.
-
-
Manuscript Structure & Figure Organization:
-
The Results section often describes data in a way that would be more appropriate for figure legends. A clearer distinction between textual explanations and figure references would improve readability.
-
Figures should be consolidated into 6 or 7 rather than the current extensive fragmentation. Many figures appear to depict replications of similar experiments with different methodologies making them more appropriate for a figure.
-
-
Quantification and Referencing Issues:
-
The quantification in Figure 14 appears inaccurate and should be reassessed to ensure proper representation of the data.
-
There is a reference to Figure 15 in the text, but no such figure is included in the manuscript. This should be corrected.
-
-
Tolerability and Long-Term Exposure Considerations:
-
The tolerability of 1g in mice should be addressed, particularly concerning potential toxicity and effective dosing ranges.
-
Time points after treatment should be more explicitly discussed, particularly in NIH3T3 cells, to assess potential toxicity in normal cells. Short-term exposure alone does not adequately reflect long-term cellular responses.
-
-
Analysis of Microtubule-Associated Proteins:
-
One would expect that the effect of this molecule would be extensively studied in the manuscript, including checking for the expression and phosphorylation status of important Microtubule-Associated Proteins such as Tau, MAP2, and stathmin. Assessing these markers would provide deeper mechanistic insights into how 1g interacts with the microtubule network.
-
Minor Revisions:
-
Formatting and Terminology Consistency:
-
Ensure consistent terminology and formatting throughout the manuscript to enhance clarity and professionalism.
-
Standardize figure labeling and legends to ensure clarity and uniformity.
-
-
Language and Readability:
-
Some sections are overly technical and would benefit from clearer phrasing to ensure accessibility to a broader scientific audience.
-
Grammar and sentence structure should be reviewed for fluency and coherence.
-
Conclusion:
The manuscript presents significant findings that contribute to the understanding of glioblastoma cell mechanics and potential therapeutic interventions. However, addressing the concerns regarding structure, data presentation, and experimental justification will enhance its clarity and impact. A revised version incorporating these suggestions would be well-positioned for publication consideration.
Author Response
Answer to Ref 1
Strengths of the Manuscript:
Scientific Merit:
The study presents a novel benzodiazepine-derived molecule (1g) that potentially affects glioblastoma cell proliferation and migration, contributing valuable insights into glioblastoma treatment.
Comprehensive Methodology:
The use of multiple experimental techniques, including micropipette aspiration, traction force microscopy, and 3D spheroid invasion models, provides a robust investigation of the effects of 1g.
Mechanistic Insights:
The manuscript effectively discusses the potential mechanisms of action, highlighting the role of microtubule-associated proteins (MAPs) and the impact of 1g on cellular biomechanics.
Clinical Relevance:
The discussion regarding the blood-brain barrier penetration of 1g suggests potential translational applications.
We thank the referee for appreciating the work and for the advice aimed at improving the presentation and clarity of the work
Major Revisions:
Introduction Clarity:
The end of the introduction contains excessive details regarding the study's results. While it is important to summarize the significance of the research, these findings should be relocated to the Discussion or Results section to enhance the logical flow.
Also following the suggestions of another referee, we have removed from the Introduction most of the detailed statements regarding the results obtained. The Introduction section now appears shorter and more focused on the presentation of the topic and on a brief preview of the main results we obtained with this investigation.
Selection of 15-20µM Concentration:
The manuscript lacks an IC50 curve for 1g, making it unclear why the authors selected 15-20µM. The inclusion of such data would strengthen the justification for the chosen concentration range.
According to the reviewer's suggestion, we added, in the supplementary material file, the D/R of 1g on U87MG at different time points with the respective IC50
Manuscript Structure & Figure Organization:
The Results section often describes data in a way that would be more appropriate for figure legends. A clearer distinction between textual explanations and figure references would improve readability.
We moved some technical details from the text to the Figure legends. For example, we inserted in the legend to Figure 2 the details of the Matrigel matrix; in the legend to Figure 3 we inserted the discussion about the mechanical model exploited to describe the experiment with the Micropipette Aspiration Technique; in the legend to Figure 7 we inserted some details about the data collection to perform the MSD analysis.
Figures should be consolidated into 6 or 7 rather than the current extensive fragmentation. Many figures appear to depict replications of similar experiments with different methodologies making them more appropriate for a figure.
Following the Referee’s suggestion, considering that Figure 2 can be considered as a high magnification of Figure 1, we grouped Figure 1 with Figure 2. Figure 5 and 6 can be grouped into just one Figure considering that Figure 5 deals with the polarity loss of cells exposed to 1g and how they are flattened on the surface. Accordingly, following the Referee’s suggestion we grouped the two images and now they are Figure 4a and Figure 4b. Figure 7, 8 and 9 of the submitted manuscript are all involved with problems in cell duplication in the presence of 1g and they have now been grouped in just one Figure (now Figure 5). We believe that the grouping of figures cannot be improved further. The new Figures are reported below(please see the attachment):
Figure 1 Figure 4
Figure 5
Quantification and Referencing Issues:
The quantification in Figure 14 appears inaccurate and should be reassessed to ensure proper representation of the data.
We thank the referee for making us realize that we had made a mistake in the presentation of the Western Blot figure. Indeed, there was a mistake in the blot panel assembly; one blot (vinculin) was acquired in a mirrored way with respect to the right one (see below). Now the figure is in the correct form.
New Figure Old Figure
There is a reference to Figure 15 in the text, but no such figure is included in the manuscript. This should be corrected.
We removed the reference to Figure 15
Tolerability and Long-Term Exposure Considerations:
The tolerability of 1g in mice should be addressed, particularly concerning potential toxicity and effective dosing ranges.
1g is a relatively new derivative coming from a lead compound called GYKI, which is an AMPA antagonist. Up to now, 1g has not been tested in-vivo. Our group focused the experimental research on the ability of the compound to interfere with cancer growth and invasion in-vitro in different cancer cell models. Of course, we agree with the referee that the toxicity is an issue that should be addressed in animal models, but at this moment, it is equally important to have a deeper and complete knowledge about the molecular mechanism of action of the compound.
Time points after treatment should be more explicitly discussed, particularly in NIH3T3 cells, to assess potential toxicity in normal cells. Short-term exposure alone does not adequately reflect long-term cellular responses.
1g did not significantly change the cell viability of NIH-3T3. To better describe this result, according to the reviewer's suggestion, we have added, in the supplementary material (Figure S3b), the D/R of 1g on NIH-3T3 after 48 h of incubation time.
Analysis of Microtubule-Associated Proteins:
One would expect that the effect of this molecule would be extensively studied in the manuscript, including checking for the expression and phosphorylation status of important Microtubule-Associated Proteins such as Tau, MAP2, and stathmin. Assessing these markers would provide deeper mechanistic insights into how 1g interacts with the microtubule network.
Considering previously published results and the results of this investigation, we suggested that 1g could affect some MAP. MAPs can be classified according to the microtubule interaction region (lattice proteins, protein binding to the plus or minus-end) or their function (stabilizing, destabilizing, capping or motor proteins). The referee is right that, especially in the case of brain tumors, Tau, a microtubule lattice binding protein which typically stabilizes microtubules and is dowregulated in glioblastoma, could be a relevant MAP to be investigated and, indeed, it will be analysed in future investigations. Other MAPS, such as ATIP3, stabilize microtubules and could favour cancer cell migration, although they have not been studied in the cases of neuron-related tumors. Their deficiency in cancer cells has been found to enhance microtubule dynamics favouring migration and invasion. In general, the possibility of targeting MAPs to fight tumors is quite interesting due to the possibility of a decreased toxicity with respect to drugs directly targeting microtubules such as taxols and vinca. Moreover, the altered expression of some MAPs correlates with the efficacy of microtubule targeting drugs and it is prognostic of the evolution of the tumor. Here, we have some pieces of evidence that 1g affects the cortical cytoskeleton and we concentrated on ERM proteins that are able to interact with microtubules near the cortical layer and for which we found an almost instantaneous increase of the phosphorylated form once cells were exposed to 1g. On the other side, we also concentrated on the role of Kif11, a kinesin that interact with microtubules during the mitotic process. This analysis was performed because a previous work in the literature had found a similar morphological behaviour in the case of glioma cells treated with ispinesib (which specifically targets Kif11). However, in the next future, we will explore other possible targets including Tau, MAP2, and stathmin.
In the Introduction section we introduced some comments on the possibility of targeting MAPs to fight tumors.
Minor Revisions:
Formatting and Terminology Consistency:
Ensure consistent terminology and formatting throughout the manuscript to enhance clarity and professionalism.
Standardize figure labeling and legends to ensure clarity and uniformity.
All the Figure labelling are now uniform and the presentation of the scale bar is the same for all Figures. We have also standardized the terminology throughout the text.
Language and Readability:
Some sections are overly technical and would benefit from clearer phrasing to ensure accessibility to a broader scientific audience.
Grammar and sentence structure should be reviewed for fluency and coherence.
We removed from the Results section unnecessary details about the methods and we went throughout all the manuscript trying to improve the grammar and the readability of the manuscript. We also included a few sentences to explain some technical details that might be too specific for a broad audience, such as for example the explanation of what is a non-linear elastic behavior of the matrix (the matrix effective Young Modulus changes, specifically increasing, upon an increase of the deformation).
Conclusion:
The manuscript presents significant findings that contribute to the understanding of glioblastoma cell mechanics and potential therapeutic interventions. However, addressing the concerns regarding structure, data presentation, and experimental justification will enhance its clarity and impact. A revised version incorporating these suggestions would be well-positioned for publication consideration.
We thank the reviewer for appreciating our work and we think that, after the suggested changes, the manuscript has improved.

Reviewer 2 Report
Comments and Suggestions for Authors
To interfere with the high invasive potential of glioblastoma-astrocytoma cells, a novel benzodiazepine-derived molecule was applied in this work. This work contains some useful information and needs some modifications.
1. Why define the molecule as “1g”? It should be defined by the name of compound.
2. The writing of whole manuscript should be improved. Avoid use too many words such as “we ...”. Meanwhile, the length of this article should be reduced by retaining the most important results and findings.
3. In the abstract, the authors have paid their attention on introducing the experiment protocols of this work. However, the results of this work are only simply described. So abstract must be revised by simply introducing the experimental design but describing the experimental results in details.
4. In the introduction, existing drugs that have been used for selectively targeting MAPs induced alteration of microtubules dynamics but preventing their depolymerization are not introduced. What are the advantages of the new compound used in this work in comparison with the reported ones? The results of this work such as “We found that 1g is able to reduce the expansion of cell spheroids in Matrigel® matrices and strongly affects (by decreasing) collective contractility. ”, “we found that, in the short term, this molecule affects cell morphology in the short term by removing cell polarity in a cell-specific manner on U87MG cells and not on fibroblast (NIH3T3) cells”, “We found an increased stiffness for single cells exposed to 1g for 24 h (cells not in mitosis and with a round shape, i.e. cells in interphase probably after an attempt of mitosis) and a decreased stiffness and surface tension for spheroids, where the mechanical properties strongly depend on the cell/cell interactions and on the extracellular matrix”, “We suggest that 1g, at the tested concentrations, acts by altering microtubule dynamics without inducing depolymerization”, “We also found pieces of evidence suggesting an interaction of the molecule at issue with the microtubule/plasma membrane attachment sites causing both a loss of polarity and an aberrant mitotic fuse formation due to the disorganization of the astral microtubules” should not be introduced in the introduction. So the introduction must be revised by clearly introducing the background and simply introduce the aim and content of this work.
5. In Materials and Methods, the preparation process and the structural characteristics of the new compound are not introduced. Many sections lack references.
6. In results, why “35 h and 68 h” are selected to observe U87MG spheroid expansion? The scale bar is not presented in most pictures in Fig. 1b, Fig. 3, Fig. 5 and Fig. 7. The morphology of the cells is not clearly shown in Fig. 2. High definition SEM can be used to observe the morphology of the cells. The authors should avoid introducing too much background information in the results. Just simply introduce the results and reduce the length of this article. Change “2.1.3. Micropipette Aspiration Technique (MAT) shows that 1g decreases surface tension of U87MG spheroids” into “1g decreases surface tension of U87MG spheroids”. In section 2.1.3, the authors should retain the most important data/results and delete unnecessary equations. I suggest adding a scheme to illustrate the mechanism of action of the new compound on U87MG and GBM cells.
Comments on the Quality of English Language
The English could be improved to more clearly express the research.
Author Response
Answer to Reviewer 2
Why define the molecule as “1g”? It should be defined by the name of compound.
According to the reviewer's suggestion, we added, in the abstract and the introduction section, the molecule’s chemical name: “1-(4-amino-3,5-dimethylphenyl)-3,5-dihydro-7,8-ethylenedioxy-4h2,3-benzodiazepin-4-one, named 1g”.
The writing of whole manuscript should be improved. Avoid use too many words such as “we ...”. Meanwhile, the length of this article should be reduced by retaining the most important results and findings.
We went throughout the entire manuscript improving the English writing. At the same time, we reduced the length of the manuscript as much as possible. Considering the different experimental methods we used and the necessary explanation of these methods, the manuscript is probably longer than a typical work, but we think that now all parts are relevant to fully understand our experiments and our message.
In the abstract, the authors have paid their attention on introducing the experiment protocols of this work. However, the results of this work are only simply described. So abstract must be revised by simply introducing the experimental design but describing the experimental results in details.
We modified the abstract by focusing on the experimental results we obtained and we postponed the details of the experimental methods to the specific section.
In the introduction, existing drugs that have been used for selectively targeting MAPs induced alteration of microtubules dynamics but preventing their depolymerization are not introduced. What are the advantages of the new compound used in this work in comparison with the reported ones?
We inserted in the Introduction section a few comments on the possibility of targeting MAPs to fight cancer. At the moment, there are not many drugs targeting MAPS in cancer disease, especially against GBM. Currently, drugs that target MAPs are in the drug design process, with some exceptions. Up to now, the existent MAP target drugs, under clinical phase 1-2, are the quinazoline-based kinesin inhibitors such as Ispinesib, which arrests cancer cell division in patients with malignant melanoma. Others MAPs-targeting drugs such as the katanin stabilizing agents GRC0321 and compound 20b, are studied pre-clinically, on NSCLC, but not in GBM. This is mostly due to the difficulty of overcoming the BBB. In this view, 1g, thanks to its lipohylic character and being a derivative of a benzodiazepine, could pass easily through the BBB. As far as the microtubule destabilizing and stabilizing agents such as Taxane and Vinca, they have been used, with success, as anticancer drugs for a long time. However, these compounds are known to cause resistance and unspecific cell death, which on the one hand can be acceptable, in terms of benefit-risk evaluation in cancer diseases, but, on the other hand the side effects are serious. Differently, 1g acts more as cytostatic. From a general point of view, the possibility of having a compound able to block both mitotic activity and cell migration in high proliferative cells, without affecting cell death, would be an additional tool in a complex pathology such as cancer. In addition, recently, it has been shown that taxane rechallenge in early metastatic breast cancer relapse resulted in a worse progression at free survival (Vasseur A. et al. Breast 2022), highlighting the importance of new molecules possessing a different mechanism of action.
The results of this work such as “We found that 1g is able to reduce the expansion of cell spheroids in Matrigel® matrices and strongly affects (by decreasing) collective contractility. ”, “we found that, in the short term, this molecule affects cell morphology in the short term by removing cell polarity in a cell-specific manner on U87MG cells and not on fibroblast (NIH3T3) cells”, “We found an increased stiffness for single cells exposed to 1g for 24 h (cells not in mitosis and with a round shape, i.e. cells in interphase probably after an attempt of mitosis) and a decreased stiffness and surface tension for spheroids, where the mechanical properties strongly depend on the cell/cell interactions and on the extracellular matrix”, “We suggest that 1g, at the tested concentrations, acts by altering microtubule dynamics without inducing depolymerization”, “We also found pieces of evidence suggesting an interaction of the molecule at issue with the microtubule/plasma membrane attachment sites causing both a loss of polarity and an aberrant mitotic fuse formation due to the disorganization of the astral microtubules” should not be introduced in the introduction. So the introduction must be revised by clearly introducing the background and simply introduce the aim and content of this work.
We removed all the cited sentences from the Introduction section and we inserted them in the Results section
In Materials and Methods, the preparation process and the structural characteristics of the new compound are not introduced. Many sections lack references.
According to the reviewer's suggestion, we added the following references related to the synthesis of the 1g molecule in the Materials and Methods section:
- Micale, S. Colleoni, G. Postorino, A. Pellicanò, M. Zappalà, J. Lazzaro, Structure-activity study of 2,3-benzodiazepin-4-ones noncompetitive AMPAR antagonists: identification of the 1-(4amino-3-methylphenyl)-3,5-dihydro-7,8-ethylenedioxy-4H-2,3-benzodiazepin-4-one as neuroprotective agent. Bioorg. Med. Chem. 1 (2008) 2200-2211.
- Zappalà, A. Pellicanò, N. Micale, F.S. Menniti, G. Ferreri, G. De Sarro, New 7,8ethylenedioxy-2,3-benzodiazepines as non-competitive AMPA receptor antagonists. Bioorg. Med. Chem. Lett. 1 (2006) 167-170.
- Grasso, G. De Sarro, A. De Sarro, N. Micale, M. Zappalà, G. Puia, Synthesis and anticonvulsant activity of novel and potent 2,3-benzodiazepine AMPA/kainate receptor antagonists. J. Med. Chem. 21 (1999) 4414-4421.
In results, why “35 h and 68 h” are selected to observe U87MG spheroid expansion? The scale bar is not presented in most pictures in Fig. 1b, Fig. 3, Fig. 5 and Fig. 7.
There is no specific reason for the selected time points. In the specific case of 35 h and 68 h they simply represent the total time of the experiment and a point at half-total-time.
In many cases, when more than one microscopy image is present in the same Figure, the scale bar is reported in just one image and it is assumed that it the same for all the images. Anyway, we uniformed the representation of the scale bar in all the Figures and we specified in the Figure legend that it is the same for more than one image.
The morphology of the cells is not clearly shown in Fig. 2. High definition SEM can be used to observe the morphology of the cells.
The images reported in Figure 2 (now Figure 1c) refer to a spheroid embedded in matrigel gel and have been acquired with a low magnification objective. Being a 3D sample it is not simple to prepare it in order to be observed by SEM. Anyway, the different cell morphology in the two cases is quite evident even if it is mainly qualitative.
The authors should avoid introducing too much background information in the results. Just simply introduce the results and reduce the length of this article.
Following also a comment from another Referee, we have removed unnecessary details on methods and background from the Results section and have tried to reduce the length of the manuscript as much as possible. We also went throughout all the manuscript trying to improve the grammar and the readability of the manuscript.
Change “2.1.3. Micropipette Aspiration Technique (MAT) shows that 1g decreases surface tension of U87MG spheroids” into “1g decreases surface tension of U87MG spheroids”.
We changed the title of the section according to the Referee’s suggestion
In section 2.1.3, the authors should retain the most important data/results and delete unnecessary equations.
We moved most of the equations to the Methods section.
I suggest adding a scheme to illustrate the mechanism of action of the new compound on U87MG and GBM cells.
Thanks to the reviewer for the suggestion. However, the graphical abstract already summarizes what we can state about the mechanism of action of 1g and we think that another scheme would not add important details.
Reviewer 3 Report
Comments and Suggestions for Authors
"A novel benzodiazepine-derived molecule interferes with the bio-mechanical properties of glioblastoma-astrocytoma cells altering their proliferation and migration" is an interesting study and has potential applications, but there are the following concerns that need to be addressed before publication:
1. Authors are suggested to remove the word "novel" from the topic and can discuss the novelty in the abstract.
2. Authors should add a few lines of perspectives at the end of the abstract.
3. Correct the heading 2.1.1.
4. Authors should add some future applications in the conclusion section.
Author Response
Answer to Reviewer 3
A novel benzodiazepine-derived molecule interferes with the bio-mechanical properties of glioblastoma-astrocytoma cells altering their proliferation and migration" is an interesting study and has potential applications, but there are the following concerns that need to be addressed before publication:
1. Authors are suggested to remove the word "novel" from the topic and can discuss the novelty in the abstract.
We removed the word novel from the title and we introduced the novelty in the abstract
Authors should add a few lines of perspectives at the end of the abstract. Authors should add some future applications in the conclusion section.
According to the reviewer's suggestion, we added some future perspectives in the conclusion section: “Although this research is focused on in-vitro models and further studies are necessary to identify and understand the specific molecular pathways involved in the activity of this novel 2,3-benzodiazepine derivative, the compound 1g has already demonstrated promising pharmacological characteristics. These features make it a strong candidate for future research and development within the field of cancer research. Moreover, the potential applications of compound 1g may extend into cancer therapy itself, offering hope for new treatment options. As the scientific community continues to explore the intricate mechanisms of this compound, it is essential to investigate its efficacy and safety further, paving the way for advancements in cancer treatment strategies.”
Correct the heading 2.1.1.
We changed the headings throughout all the document
Round 2
Reviewer 2 Report
Comments and Suggestions for Authors
The revised manuscript can be accepted.